# INCREASE AND CONQUER: TRAINING GRAPH NEURAL NETWORKS ON GROWING GRAPHS

## ABSTRACT

Graph neural networks (GNNs) use graph convolutions to exploit network invariances and learn meaningful features from network data. However, on large-scale graphs convolutions incur in high computational cost, leading to scalability limitations. Leveraging the graphon — the limit object of a graph — in this paper we consider the problem of learning a graphon neural network (WNN) — the limit object of a GNN — by training GNNs on graphs sampled Bernoulli from the graphon. Under smoothness conditions, we show that: (i) the expected distance between the learning steps on the GNN and on the WNN decreases asymptotically with the size of the graph, and (ii) when training on a sequence of growing graphs, gradient descent follows the learning direction of the WNN. Inspired by these results, we propose a novel algorithm to learn GNNs on large-scale graphs that, starting from a moderate number of nodes, successively increases the size of the graph during training. This algorithm is benchmarked on both a recommendation system and a decentralized control problem where it is shown to retain comparable performance to its large-scale counterpart at a reduced computational cost.

## 1   INTRODUCTION

Graph Neural Networks (GNNs) are deep convolutional architectures formed by a succession of layers where each layer composes a graph convolution and a pointwise nonlinearity (Wu et al., 2021; Zhou et al., 2020). Tailored to network data, GNNs have been used in a variety of applications such as recommendation systems (Fan et al., 2019; Tan et al., 2020; Ying et al., 2018; Schlichtkrull et al., 2018; Ruiz et al., 2019a) and Markov chains (Qu et al., 2019; Ruiz et al., 2019b; Li et al., 2015), and fields such as biology (Fout et al., 2017; Duvenaud et al., 2015; Gilmer et al., 2017; Chen et al., 2020) and robotics (Qi et al., 2018; Gama & Sojoudi, 2021; Li et al., 2019). Their success in these fields and applications provides ample empirical evidence of the ability of GNNs to generalize to unseen data. More recently, their successful performance has also been justified by theoretical works showing that GNNs are invariant to relabelings (Chen et al., 2019; Keriven & Peyré, 2019), stable to graph perturbations (Gama et al., 2020) and transferable across graphs (Ruiz et al., 2020a).

One of the most important features of a GNN is that, because the linear operation is a graph convolution, its number of parameters does not depend on the number of nodes of the graph. In theory, this means that GNNs can be trained on graphs of any size. In practice, however, if the graph has large number of nodes training the GNN is costly because computing graph convolutions involves large matrix operations. While this issue could be mitigated by transferability — training the GNN on a smaller graph to execute on the large graph —, this approach does not give any guarantees on the distance between the optimal solutions on the small and on the large graph. In other words, when executing the GNN on the target graph we do not know if its error will be dominated by the transferability error or by the generalization error from training.

In this paper, we address the computational burden of training a GNN on a large graph by progressively increasing the size of the network. We consider the limit problem of learning an "optimal" neural network for a graphon, which is both a graph limit and a random graph model (Lovász, 2012). We postulate that, because sequences of graphs sampled from the graphon converge to it, the so-called graphon neural network (Ruiz et al., 2020a) can be learned by sampling graphs of growing size and training a GNN on these graphs (Algorithm 1). We prove that this is true in two steps. In

Theorem 1, we bound the expected distance between the gradient descent steps on the GNN and on the graphon neural network by a term that decreases asymptotically with the size of the graph. A consequence of this bias bound is that it allows us to quantify the trade-off between a more accurate gradient and one that could be obtained with less computational power. We then use this theorem to prove our main result in Theorem 2, which is stated in simplified form below.

**Theorem** (Graphon neural network learning, informal) Let $\mathbf{W}$ be a graphon and let $\{\mathbf{G}_n\}$ be a sequence of growing graphs sampled from $\mathbf{W}$. Consider the graphon neural network $\mathbf{\Phi}(\mathbf{W})$ and assume that it is learned by training the GNN $\mathbf{\Phi}(\mathbf{G}_n)$ with loss function $\ell(\mathbf{y}_n, \mathbf{\Phi}(\mathbf{G}_n))$ on the sequence $\{\mathbf{G}_n\}$. Over a finite number of training steps, we obtain

$$\|\nabla\ell(Y, \mathbf{\Phi}(\mathbf{W})\| \leq \epsilon \text{ with probability } 1.$$

The most important implication of this result is that the learning iterates computed in the sequence of growing graphs follow the direction of the graphon gradient up to a small ball, which provides theoretical validation to our cost efficient training methodology. We also validate our algorithm in two numerical experiments. In the first, we learn a GNN-based recommendation system on increasingly large subnetworks of a movie similarity graph, and compare it with the recommendation system trained on the full graph. In the second, we consider the problem of flocking and train GNNs to learn the actions agents need to take to flock. We compare the results obtained when progressively increasing the number of agents during training and when training directly on the target graph.

## 2 RELATED WORK

GNNs are data processing architectures that follow from the seminal works in the areas of deep learning applied to graph theory (Bruna et al., 2013; Defferrard et al., 2016; Gori et al., 2005; Lu & Getoor, 2003). They have been successfully used in a wide variety of statistical learning problems (Kipf & Welling, 2016; Scarselli et al., 2018), where their good performance is generally attributed to the fact that they exploit invariances present in network data (Maron et al., 2019; Gama et al., 2018; Chami et al., 2021).

More recently, a number of works show that GNNs can be transferred across graphs of different sizes (Ruiz et al., 2020a; Levie et al., 2019; Keriven et al., 2020). Specifically, (Ruiz et al., 2020a) leverages graphons to define families of graphs within which GNNs can be transferred with an error bound that decreases asymptotically with the size of the graph. The papers by Levie et al. (2019) and Keriven et al. (2020) offer similar results by considering the graph limit to be a generic topological space and a random graph model respectively. In this paper, we use an extension of the transferability bound derived in Ruiz et al. (2020a) to propose a novel learning algorithm for GNNs.

## 3 PRELIMINARY DEFINITIONS

### 3.1 GRAPH NEURAL NETWORKS

Graph neural networks exploit graph symmetries to extract meaningful information from network data (Ruiz et al., 2020c; Gama et al., 2020). Graphs are represented as triplets $\mathbf{G}_n = (\mathcal{V}, \mathcal{E}, W)$, where $\mathcal{V}$, $|\mathcal{V}| = n$, is the set of nodes, $\mathcal{E} \subseteq \mathcal{V} \times \mathcal{V}$ is the set of edges and $W : \mathcal{E} \to \mathbb{R}$ is a map assigning weights to each edge. The graph $\mathbf{G}_n$ can also be represented by the graph shift operator (GSO) $\mathbf{S} \in \mathbb{R}^{n \times n}$, a square matrix that respects the sparsity of the graph. Examples of GSOs include the adjacency matrix $\mathbf{A}$, the graph Laplacian $\mathbf{L} = \text{diag}(\mathbf{A}\mathbf{1}) - \mathbf{A}$ and their normalized counterparts Gama et al. (2018). In this paper we consider the graph $\mathbf{G}_n$ to be undirected and fix $\mathbf{S} = \mathbf{A}/n$.

Graph data is represented in the form of graph signals. A graph signal $\mathbf{x} = [x_1, \ldots, x_n]^T \in \mathbb{R}^n$ is a vector whose $i$-th component corresponds to the information present at the $i$-th node of graph $\mathbf{G}_n$. A basic data aggregation operation can be defined by applying the GSO $\mathbf{S}$ to graph signals $\mathbf{x}$. The resulting signal $\mathbf{z} = \mathbf{S}\mathbf{x}$ is such that the data at node $i$ is a weighted average of the information in the 1-hop neighborhood of $i$, $z_i = \sum_{j \in \mathcal{N}_i} [\mathbf{S}]_{ij} x_j$ where $\mathcal{N}_i = \{j \mid [\mathbf{S}]_{ij} \neq 0\}$. Information coming from further neighborhoods can be aggregated by successive applications of the GSO, also called *shifts*. Using this notion of shift, graph convolutions are defined by weighting the contribution of each successive application of $\mathbf{S}$ to define a polynomial in the GSO. Explicitly, the graph

convolutional filter with coefficients $\mathbf{h} = [h_0, \ldots, h_{K-1}]$ is given by

$$\mathbf{y} = \mathbf{h} *_{\mathbf{S}} \mathbf{x} = \sum_{k=0}^{K-1} h_k \mathbf{S}^k \mathbf{x} \tag{1}$$

where $*_{\mathbf{S}}$ denotes the convolution operation with GSO $\mathbf{S}$.

Since the adjacency matrix of an undirected graph is always symmetric, the GSO admits an eigen-decomposition $\mathbf{S} = \mathbf{V}\mathbf{\Lambda}\mathbf{V}^H$. The columns of $\mathbf{V}$ are the graph eigenvectors and the diagonal elements of $\mathbf{\Lambda}$ are the graph eigenvalues, which take values between $-1$ and $1$ and are ordered as $-1 \leq \lambda_{-1} \leq \lambda_{-2} \leq \ldots \leq 0 \leq \ldots \leq \lambda_2 \leq \lambda_1 \leq 1$. Since the eigenvectors of $\mathbf{S}$ form an orthonormal basis of $\mathbb{R}^n$, we can project (1) onto this basis to obtain the spectral representation of the graph convolution, which is given by

$$h(\lambda) = \sum_{k=0}^{K-1} h_k \lambda^k. \tag{2}$$

Note that (2) only depends on the $h_k$ and on the eigenvalues of the GSO. Hence, as a consequence of the Cayley-Hamilton theorem, convolutional filters may be used to represent any graph filter with spectral representation $h(\lambda) = f(\lambda)$ where $f$ is analytic (Strang, 1976).

Graph neural networks are layered architectures where each layer consists of a graph convolution followed by a pointwise nonlinearity $\rho$, and where each layer's output is the input to the following layer. At layer $l$, a GNN can output multiple features $\mathbf{x}_l^f$, $1 \leq f \leq F_l$ which we stack in a matrix $\mathbf{X}_l = [\mathbf{x}_l^1, \ldots, \mathbf{x}_l^{F_l}] \in \mathbb{R}^{n \times F_l}$. Each column of the feature matrix is the value of the graph signal at feature $f$. To map the $F_{l-1}$ features coming from layer $l-1$ into $F_l$ features, $F_{l-1} \times F_l$ convolutions need to be computed, one per input-output feature pair. Stacking their weights in $K$ matrices $\mathbf{H}_{lk} \in \mathbb{R}^{F_{l-1} \times F_l}$, we write the $l$-th layer of the GNN as

$$\mathbf{X}_l = \rho \left( \sum_{k=0}^{K-1} \mathbf{S}^k \mathbf{X}_{l-1} \mathbf{H}_{lk} \right). \tag{3}$$

In an $L$-layer GNN, the operation in (3) is cascaded $L$ times to obtain the GNN output $\mathbf{Y} = \mathbf{X}_L$. At the first layer, the GNN input is given by $\mathbf{X}_0 = \mathbf{X} \in \mathbb{R}^{n \times F_0}$. In this paper we assume $F_0 = F_L = 1$ so that $\mathbf{Y} = \mathbf{y}$ and $\mathbf{X} = \mathbf{x}$. A more concise representation of this GNN can be obtained by grouping all learnable parameters $\mathbf{H}_{lk}$ in a tensor $\mathcal{H} = \{\mathbf{H}_{lk}\}_{l,k}$ and defining the map $\mathbf{y} = \mathbf{\Phi}(\mathbf{x}; \mathcal{H}, \mathbf{S})$. Due to the polynomial nature of the graph convolution, the dimensions of the learnable parameter tensor $\mathcal{H}$ are independent from the size of the graph ($K$ is typically much smaller than $n$). Ergo, a GNN trained on a graph $\mathbf{G}_n$ can be deployed on a network $\mathbf{G}_m$ with $m \neq n$.

## 3.2 GRAPHON INFORMATION PROCESSING

A graphon is a bounded, symmetric, and measurable function $\mathbf{W} : [0,1]^2 \to [0,1]$ which has two theoretical interpretations — it is both a graph limit and a generative model for graphs. In the first interpretation, sequences of dense graphs converge to a graphon in the sense that the densities of adjacency-preserving graph motifs converge to the same densities on the graphon Lovász (2012). In the second, graphs can be generated from a graphon by sampling points $u_i, u_j$ from the unit interval and either assigning weight $\mathbf{W}(u_i, u_j)$ to edges $(i,j)$, or sampling edges $(i,j)$ with probability $\mathbf{W}(u_i, u_j)$. In this paper, we focus on stochastic graphs $\mathbf{G}_n$ where the points $u_i$ are defined as $u_i = (i-1)/n$ for $1 \leq i \leq n$ and where the adjacency matrix $\mathbf{S}_n$ is sampled from $\mathbf{W}$ as

$$[\mathbf{S}_n]_{ij} \sim \text{Bernoulli}(\mathbf{W}(u_i, u_j)). \tag{4}$$

Sequences of graphs generated in this way can be shown to converge to $\mathbf{W}$ with probability one (Lovász, 2012)[Chapter 11].

In practice, the two theoretical interpretations of a graphon allow thinking of it as an identifying object for a *family* of graphs of different sizes that are structurally similar. Hence, given a network we can use its family's identifying graphon as a continuous proxy for the graph. This is beneficial because it is typically easier to operate in continuous than in discrete domains, even more so if the network is large. We will leverage these ideas to consider graphon data and graphon neural networks as proxies for graph data and GNNs supported on graphs of arbitrary size.

### 3.2.1 GRAPHON NEURAL NETWORKS

Graphon data is defined as functions $X \in L^2([0,1])$. Analogously to graph data, graphon data can be diffused by application of a linear integral operator parametrized by $\mathbf{W}$ and defined as

$$T_{\mathbf{W}} X(v) = \int_0^1 \mathbf{W}(u,v) X(u) du. \tag{5}$$

The operator $T_{\mathbf{W}}$ is called graphon shift operator (WSO).

The graphon convolution is defined as a weighted sum of successive applications of the WSO. Explicitly, the graphon convolutional filter with coefficients $\mathbf{h} = [h_0, \ldots, h_{K-1}]$ is given by

$$Y = \mathbf{h} *_{\mathbf{W}} X = \sum_{k=0}^{K-1} h_k (T_{\mathbf{W}}^{(k)} X)(v) \quad \text{with}$$

$$(T_{\mathbf{W}}^{(k)} X)(v) = \int_0^1 \mathbf{W}(u,v)(T_{\mathbf{W}}^{(k-1)} X)(u) du \tag{6}$$

where $T_{\mathbf{W}}^{(0)} = \mathbf{I}$ is the identity (Ruiz et al., 2020b). Since $\mathbf{W}$ is bounded and symmetric, $T_{\mathbf{W}}$ is a self-adjoint Hilbert-Schmidt operator (Lax, 2002). Hence, $\mathbf{W}$ can be written as $\mathbf{W}(u,v) = \sum_{i \in \mathbb{Z} \setminus \{0\}} \lambda_i \varphi_i(u) \varphi_i(v)$ where $\lambda_i$ are the graphon eigenvalues and $\varphi_i$ the graphon eigenfunctions. The eigenvalues have magnitude at most one and are ordered as $-1 \leq \lambda_{-1} \leq \lambda_{-2} \leq \ldots \leq 0 \leq \ldots \lambda_2 \leq \lambda_1 \leq 1$. The eigenfunctions form an orthonormal basis of $L^2([0,1])$. Projecting the filter (1) onto this basis, we see that the graphon convolution admits a spectral representation given by

$$h(\lambda) = \sum_{k=0}^{K-1} h_k \lambda^k. \tag{7}$$

Like its graph counterpart, this spectral representation only depends on the eigenvalues of the graphon.

Graphon neural networks (WNNs) are the extension of GNNs to graphon data. In the WNN, each layer consists of a bank of graphon convolutions (6) followed by a nonlinearity $\rho$. Assuming that layer $l$ maps $F_{l-1}$ features into $F_l$ features, the parameters of the $F_{l-1} \times F_l$ convolutions (6) can be stacked into $K$ matrices $\{\mathbf{H}_{lk}\} \in \mathbb{R}^{F_{l-1} \times F_l}$. This allows writing the $f$th feature at layer $l$ as

$$X_l^f = \rho \left( \sum_{g=1}^{F_{l-1}} \sum_{k=1}^{K-1} (T_{\mathbf{W}}^{(k)} X_{l-1}^g)[\mathbf{H}_{lk}]_{gf} \right) \tag{8}$$

for $1 \leq f \leq F_l$. For an $L$-layer WNN, (8) is repeated for $1 \leq \ell \leq L$. The WNN output is given by $Y^f = X_L^f$, and $X_0^g$ is given by the input data $X^g$ for $1 \leq g \leq F_0$. We assume $F_L = F_0 = 1$ so that $X_L = Y$ and $X_0 = X$. A more succinct representation of this WNN is the map $Y = \mathbf{\Phi}(X; \mathcal{H}, \mathbf{W})$, where the tensor $\mathcal{H} = \{\mathbf{H}_{lk}\}_{l,k}$ groups the filter coefficients at all layers.

### 3.2.2 SAMPLING GNNS FROM WNNS

From the representation of the GNN and the WNN as maps $\mathbf{\Phi}(\mathbf{x}; \mathcal{H}, \mathbf{S})$ and $\mathbf{\Phi}(X; \mathcal{H}, \mathbf{W})$, we see that these architectures can share the same filter coefficients $\mathcal{H}$. Since graphs can be obtained from graphons as in (4), we can similarly use the WNN $\mathbf{\Phi}(X; \mathcal{H}, \mathbf{W})$ to *sample* GNNs

$$\mathbf{y}_n = \mathbf{\Phi}(\mathbf{x}_n; \mathcal{H}, \mathbf{S}_n) \text{ where } [\mathbf{S}_n]_{ij} \sim \text{Bernoulli}(\mathbf{W}(u_i, u_j))$$

$$[\mathbf{x}_n]_i = X(u_i) \tag{9}$$

i.e., the WNN can be seen as a generative model for GNNs $\mathbf{\Phi}(\mathbf{x}_n; \mathcal{H}, \mathbf{S}_n)$.

Conversely, a WNN ca be induced by a GNN. Given a GNN $\mathbf{y}_n = \mathbf{\Phi}(\mathbf{x}_n; \mathcal{H}, \mathbf{S}_n)$, the WNN induced by this GNN is defined as

$$Y_n = \mathbf{\Phi}(X_n; \mathcal{H}, \mathbf{W}_n) \text{ where } \mathbf{W}_n(u,v) = \sum_{i=1}^n \sum_{j=1}^n [\mathbf{S}_n]_{ij} \mathbb{I}(u \in I_i) \mathbb{I}(v \in I_j)$$

$$X_n(u) = \sum_{i=1}^n [\mathbf{x}_n]_i \mathbb{I}(u \in I_i) \tag{10}$$

where $\mathbb{I}$ denotes the indicator function and the intervals $I_i$ are defined as $I_i = [(i-1)/n, i/n)$ for $1 \leq i \leq n-1$ and $I_n = [(n-1)/n, 1]$. The graphon $\mathbf{W}_n$ is called the *graphon induced by the graph* $\mathbf{G}_n$ and $X_n$ and $Y_n$ are called the *graphon signals induced by the graph signals* $\mathbf{x}_n$ and $\mathbf{y}_n$.

## 4 GRAPHON EMPIRICAL LEARNING

On graphons, the statistical loss minimization (or statistical learning) problem is given by

$$\underset{\mathcal{H}}{\text{minimize}} \quad \mathbb{E}_{p(Y,X)}[\ell(Y, \mathbf{\Phi}(X; \mathcal{H}, \mathbf{W}))] \tag{11}$$

where $p(Y, X)$ is the joint distribution of the data, $\ell$ is an instantaneous loss function and $\mathbf{\Phi}(X; \mathcal{H}, \mathbf{W})$ is a function parametrized by the graphon $\mathbf{W}$ and by a set of learnable weights $\mathcal{H}$. In this paper, we consider positive loss functions $\ell : \mathbb{R} \times \mathbb{R} \to \mathbb{R}^+$. The function $\mathbf{\Phi}$ is parametrized as a graphon neural network [cf. (8)]. Because the joint probability distribution $p(Y, X)$ is unknown, we are unable to derive a closed-form solution of (11), but this problem can be approximated by the empirical risk minimization (ERM) problem over graphons.

### 4.1 GRAPHON EMPIRICAL RISK MINIMIZATION

Suppose that we have access to samples of the distribution $\mathcal{D} = \{(X^j, Y^j) \sim p(X, Y), j = 1, \ldots, |\mathcal{D}|\}$. Provided that these samples are obtained independently and that $|\mathcal{D}|$ is large enough, the statistical loss in (11) can be approximated as

$$\underset{\mathcal{H}}{\text{minimize}} \quad \sum_{j=1}^{|\mathcal{D}|} \ell(Y^j, \mathbf{\Phi}(X^j; \mathcal{H}, \mathbf{W})) \tag{12}$$

giving way to the ERM problem (Hastie et al., 2009; Shalev-Shwartz & Ben-David, 2014; Vapnik, 1999; Kearns et al., 1994). To solve this problem using local information, we could adopt some flavor of gradient descent (Goodfellow et al., 2016). The learning iteration at step $k$ is then

$$\mathcal{H}_{k+1} = \mathcal{H}_k - \eta_k \nabla_{\mathcal{H}} \ell(Y^j, \mathbf{\Phi}(X^j; \mathcal{H}_k, \mathbf{W})) \tag{13}$$

where $k = 1, 2, \ldots$ denotes the current iteration and $\eta_k \in (0, 1)$ the step size at iteration $k$.

In practice, the gradients on the right hand side of (13) cannot be computed because, being a theoretical limit object, the graphon $\mathbf{W}$ is unknown. However, we can leverage the fact that the graphon is a random graph model to approximate the gradients $\nabla_{\mathcal{H}} \ell(Y^j, \mathbf{\Phi}(X^j; \mathcal{H}, \mathbf{W}))$ by sampling stochastic graphs $\mathbf{G}_n$ with GSO $\mathbf{S}_n$ [cf. (4)] and calculating $\nabla_{\mathcal{H}} \ell(\mathbf{y}_n^j, \mathbf{\Phi}(\mathbf{x}_n^j; \mathcal{H}, \mathbf{S}_n))$, where $\mathbf{x}_n^j$ and $\mathbf{y}_n^j$ are as in (9). In this case, the graphon empirical learning step in (13) becomes

$$\mathcal{H}_{k+1} = \mathcal{H}_k - \eta_k \nabla_{\mathcal{H}} \ell(\mathbf{y}_n^j, \mathbf{\Phi}(\mathbf{x}_n^j; \mathcal{H}_k, \mathbf{S}_n)) \tag{14}$$

and we have to derive an upper bound for the expected error made when using the gradient calculated on the graph to approximate the gradient on the graphon. In what follows, we give a closed-form expression of this bound, and use it as a stepping stone to develop an Algorithm that increases $n$, the graph size, over regular intervals during the training process of the GNN. We then prove that our Algorithm converges to a neighborhood of the optimal solution for the WNN.

### 4.2 GRADIENT APPROXIMATION

To state our first convergence result, we need following definition.

**Definition 1** (Lipschitz functions). *A function* $f(u_1, u_2, \ldots, u_d)$ *is A-Lipschitz on the variables* $u_1, \ldots, u_d$ *if it satisfies* $|f(v_1, v_2, \ldots, v_d) - f(u_1, u_2, \ldots, u_d)| \leq A \sum_{i=1}^{d} |v_i - u_i|$. *If* $A = 1$, *we say that this function is normalized Lipschitz.*

We also need the following Lipschitz continuity assumptions (AS1–AS4), as well as an assumption on the size of the graph $\mathbf{S}_n$ (AS5).

**AS1.** *The graphon* $\mathbf{W}$ *and the graphon signals* $X$ *and* $Y$ *are normalized Lipschitz.*

**AS2.** *The convolutional filters* $h$ *are normalized Lipschitz and non-amplifying, i.e.,* $\|h(\lambda)\| < 1$.

---

**Algorithm 1** Increase and Conquer: Growing Graph Training

---

1: Initialize $\mathcal{H}_0, n_0$ and sample graph $\mathbf{G}_{n_0}$ from graphon $\mathbf{W}$
2: **repeat** for epochs $0, 1, \ldots$
3:     **for** $k = 1, \ldots, |\mathcal{D}|$ **do**
4:         Obtain sample $(Y, X) \sim \mathcal{D}$
5:         Construct graph signal $\mathbf{y}_n, \mathbf{x}_n$ [cf. (9)]
6:         Take learning step $\mathcal{H}_{k+1} = \mathcal{H}_k - \eta_k \nabla \ell(\mathbf{y}_n, \mathbf{\Phi}(\mathbf{x}_n; \mathcal{H}_k, \mathbf{S}_n))$
7:     **end for**
8:     Increase number of nodes $n$ and sample $\mathbf{S}_n$ Bernoulli from graphon $\mathbf{W}$ [cf. (4)]
9: **until** convergence

---

**AS3.** *The activation functions and their gradients are normalized Lipschitz, and $\rho(0) = 0$.*

**AS4.** *The loss function $\ell : \mathbb{R} \times \mathbb{R} \to \mathbb{R}^+$ and its gradient are normalized Lipschitz, and $\ell(x, x) = 0$.*

**AS5.** *For a fixed value of $\xi \in (0, 1)$, $n$ is such that $n - \log(2n/\xi)/d_{\mathbf{W}} > 2/d_{\mathbf{W}}$ where $d_{\mathbf{W}}$ denotes the maximum degree of the graphon $\mathbf{W}$, i.e., $d_{\mathbf{W}} = \max_v \int_0^1 \mathbf{W}(u, v) du$.*

AS1–AS4 are normalized Lipschitz smoothness conditions which can be relaxed by making the Lipschitz constant greater than one. AS3 holds for most typical activation functions. AS4 can be achieved by normalization and holds for most loss functions in a closed set (e.g., the hinge or mean square losses). AS5 is necessary to guarantee a $\mathcal{O}(\sqrt{\log n / n})$ rate of convergence of $\mathbf{S}_n$ to $\mathbf{W}$ (Chung & Radcliffe, 2011).

Under AS1–AS5, the following theorem shows that the expected norm of the difference between the graphon and graph gradients in (13) and (14) is bounded. The proof is deferred to the appendices.

**Theorem 1.** *Consider the ERM problem in (12) and let $\mathbf{\Phi}(X; \mathcal{H}, \mathbf{W})$ be an L-layer WNN with $F_0 = F_L = 1$, and $F_l = F$ for $1 \leq l \leq L - 1$. Let $c \in (0, 1]$ and assume that the graphon convolutions in all layers of this WNN have K filter taps [cf. (6)]. Let $\mathbf{\Phi}(\mathbf{x}_n; \mathcal{H}, \mathbf{S}_n)$ be a GNN sampled from $\mathbf{\Phi}(X; \mathcal{H}, \mathbf{W})$ as in (9). Under assumptions AS1–AS5, it holds that*

$$\mathbb{E}[\|\nabla_{\mathcal{H}} \ell(Y, \mathbf{\Phi}(X; \mathcal{H}, \mathbf{W})) - \nabla_{\mathcal{H}} \ell(Y_n, \mathbf{\Phi}(X_n; \mathcal{H}, \mathbf{W}_n))\|] \leq \gamma c + \mathcal{O}\left( \sqrt{\frac{\log(n^{3/2})}{n}} \right) \qquad (15)$$

*where $Y_n$ is the graphon signal induced by $[\mathbf{y}_n]_i = Y(u_i)$ [cf. (10)], and $\gamma$ is a constant that depends on the number of layers L, features F, and filter taps K of the GNN [cf. Definition 7 in the Appendix].*

The bound in Theorem 1 quantifies the maximum distance between the learning steps on the graph and on the graphon as a function of the size of the graph. This bound is controlled by two terms. On the one hand, we have the *approximation bound*, a term that decreases with $n$. The approximation bound is related to the approximation of $\mathbf{W}$ by $\mathbf{S}_n$. On the other, we have the *nontransferable bound*, which is constant and controlled by $c$. This term is related to a threshold that we impose on the convolutional filter $h$. Since graphons $\mathbf{W}$ have an infinite spectrum that accumulates around zero, convergence of all spectral components of the filtered data on $\mathbf{W}_n$ can only be shown in the limit of $n$ (Ruiz et al., 2020a). Hence, given that $n$ is finite we can only upper bound distances between spectral components associated with eigenvalues larger than some $c \in (0, 1]$ that we fix. Meanwhile, the perturbations associated with the other spectral components are controlled by bounding the variation of the filter response below $c$.

The bound in Theorem 1 is important because it allows us to quantify the error incurred by taking gradient steps not on the graphon, but on the graph data. Although we cannot take gradients on the function that we want to learn [cf. (13)], by measuring the ratio between the norm of the gradient of the loss on the GNN, and the difference between the two gradients, we can expect to follow the direction of the gradient on the WNN. This is instrumental for learning a meaningful solution of the graphon ERM problem (12). Nonetheless, we are only able to decrease this approximation bound down to the value of the nontransferable bound.

### 4.3 Algorithm Construction

Since the discretization error depends on the number of nodes of the graph, we can iteratively increase the graph size at every epoch. Even if a bias is introduced in the gradient we will be able to follow the gradient direction of the graphon learning problem (13). At the same time, we need to keep the computational cost of the iterates under control. Note that the norm of the gradient is larger at first, but it decreases as we approach the optimal solution. Exploiting this behavior, we may progressively reduce the bias term as iterations increase. The idea here is to keep the discretization of the gradient small so as to closely follow the gradient direction of the graphon learning problem, but without being too conservative as this would incur in high computational cost.

In practice, in the ERM problem the number of nodes that can be added is upper bounded by the available data, which is defined nodewise on the graph. Hence, we can arbitrarily decide which and how many nodes to consider for training. The novelty of Algorithm 1 is that we do not use the largest graph available in the dataset at every epoch to train the GNN. Instead, we set a minimum graph size $n_0$ and progressively increase it up to the total number of nodes. The main advantage of Algorithm 1 is thus that it allows reducing the computational cost of training without compromising GNN performance. In what follows, we will provide the conditions under which Algorithm 1 converges to a neighborhood of the optimal solution on the graphon.

### 4.4 Algorithm Convergence

We have shown that the learning step on the graphon and on the graph are close. Now, it remains to show the practical implications of Theorem 1 for obtaining the solution of the graphon ERM problem (12). If the expected difference between the gradients is small, the iterations generated by (14) will be able to follow the direction of the true gradient on the graphon learning problem. But because the distance between gradients is inversely proportional to $n$, we need to strike a balance between obtaining a good approximation and minimizing the computational cost. In Theorem 2, we show that Algorithm 1 converges if the rate at which the graph grows is chosen to satisfy the condition given in (16).

**AS6.** *The graphon neural network $\mathbf{\Phi}(X; \mathcal{H}, \mathbf{W})$ is $A_{\mathbf{\Phi}}$-Lipschitz, and its gradient $\nabla_{\mathcal{H}} \mathbf{\Phi}(X; \mathcal{H}, \mathbf{W})$ is $A_{\nabla \mathbf{\Phi}}$-Lipschitz, with respect to the parameters $\mathcal{H}$ [cf. Definition 1].*

**Theorem 2.** *Consider the ERM problem in (12) and let $\mathbf{\Phi}(X; \mathcal{H}, \mathbf{W})$ be an L-layer WNN with $F_0 = F_L = 1$, and $F_l = F$ for $1 \leq l \leq L - 1$. Let $c \in (0, 1]$, $\epsilon \in (0, 1 - A_{\nabla \mathbf{\Phi}} \eta)$, and assume that the graphon convolutions in all layers of this WNN have $K$ filter taps [cf. (6)]. Let $\mathbf{\Phi}(\mathbf{x}_n; \mathcal{H}, \mathbf{S}_n)$ be a GNN sampled from $\mathbf{\Phi}(X; \mathcal{H}, \mathbf{W})$ as in (9). Consider the iterates generated by equation (16). Under Assumptions AS1-AS6, if at each step $k$ the number of nodes $n$ verifies*

$$\gamma c + \mathcal{O}\left( \sqrt{\frac{\log(n^{3/2})}{n}} \right) < \frac{1 - A_{\nabla \ell} \eta - \epsilon}{2} \| \nabla_{\mathcal{H}} \ell(Y_n, \mathbf{\Phi}(X_n; \mathcal{H}_k, \mathbf{W}_n)) \| \tag{16}$$

*then in finite time we will achieve an iterate $k^*$ such that the coefficients $\mathcal{H}_{k^*}$ satisfy*

$$\mathbb{E}[\| \nabla_{\mathcal{H}} \ell(Y, \mathbf{\Phi}(X; \mathcal{H}_{k^*}, \mathbf{W})) \|] \leq 2\gamma c \quad \text{after at most } k^* = \mathcal{O}(1/\epsilon) \tag{17}$$

*where $\gamma$ is a constant that depends on the number of layers L, features F, and filter taps K of the GNN [cf. Definition 7 in the Appendix].*

Theorem 2 presents the conditions under which Algorithm 1 converges. Intuitively, the condition in (16) implies that the rate of decrease in the norm of the gradient of the loss function computed on the graph neural network needs to be slower than roughly $n^{-1/2}$. If the norm of the gradient does not vary between iterations, the number of nodes does not need to be increased. Note that (115) is equal to twice the bias term obtained in equation (15). We can keep increasing the number of nodes — and thus decreasing the bias — until the norm of the GNN gradient is smaller than the nontransferable constant value. Once this point is attained, there is no gain in decreasing the approximation bound any further (Ajalloeian & Stich, 2020). Recall that the constant term can be made as small as desired by tuning $c$, at the cost of decreasing the approximation bound. To conclude, observe that assuming smoothness of the GNN is a mild assumption (Scaman & Virmaux, 2018; Jordan & Dimakis, 2020; Latorre et al., 2020; Fazlyab et al., 2019; Tanielian & Biau, 2021; Du et al., 2019). A characterization of the Lipschitz constant is an interesting research question but is out of the scope of this work.

## 5 NUMERICAL RESULTS

### 5.1 RECOMMENDATION SYSTEM

We consider a social recommendation problem given by movie ratings, for which we use the Movie-Lens 100k dataset Harper & Konstan (2015). The dataset contains $100,000$ integer ratings between 1 and 5, that were collected between $U = 943$ users and $M = 1682$ movies. We consider the problem of predicting the rating that different users would give to the movie "Contact". To exploit the social connectivity of the problem, we build the movie similarity graph, by computing the pairwise correlations between the different movies in the training set Huang et al. (2018). Further details about training splits, and hyperparameter selection can be found in the supplementary material.

For the experiment, we started the GNNs with 200, and at 300 nodes and added $\{25, 50, 75, 100\}$ nodes per epochs. The total number of epochs of the GNN trained on 1000 nodes with which we compare, is given by the maximum number of epochs that the Algorithm 1 can be run adding 25 nodes per epoch. In Figure 1, we are able to see the empirical manifestation of the benefit of Algorithm 1. Namely, regardless of the number of nodes added per epoch, in all four cases, using relative RMSE as a figure of merit, we achieve a comparable performance to that of a GNNs trained on 1000 nodes for a larger number of epochs. This reinforces the fact that training on a growing number of graphs can attain similar performance, while requiring a lighter computational cost.

### 5.2 DECENTRALIZED CONTROL

In this section we consider the problem of coordinating a set of $n$ agents initially flying at random to avoid collisions, and to fly at the same velocity. Also known as flocking, at each time $t$ agent $i$ knows its own position $r_i(t) \in \mathbb{R}^2$, and speed $v_i(t) \in \mathbb{R}^2$, and reciprocally exchanges it with its neighboring nodes if a communication link exists between them. Links are govern by physical proximity between agents forming a time varying graph $\mathbf{G}_n = (\mathcal{V}, \mathcal{E})$. A communication link exists if the distance between two agents $i, j$ satisfies $r_{ij}(t) = \|r_i(t) - r_j(t)\| \leq R = 2m$. We assume that at each time $t$ the controller sets an acceleration $u_i \in [-10, 10]^2$, and that it remains constant for a time interval $T_s = 20ms$. The system dynamics are govern by, $r_i(t+1) = u_i(t)T_s^2/2 + v_i(t)T_s + r_i(t)$, $v_i(t+1) = u_i(t)T_s + v_i(t)$. To avoid the swarm of robots to reach a null common velocity, we initialize the velocities at random $\mathbf{v}(t) = [v_1(t), \ldots, v_n(t)]$, by uniformly sampling a common bias velocity $v_{BIAS} \sim \mathcal{U}[-3, 3]$, and then adding independent uniform noise $\mathcal{U}[-3, 3]$ to each agent. The initial deployment is randomly selected in a circle always verifying that the minimum distance between two agents is larger than $0.1m$. On the one hand, agents pursue a common average velocity $\bar{\mathbf{v}} = (1/n) \sum_{i=1}^{n} v_i(t)$, thus minimizing the velocity variation of the team. On the other

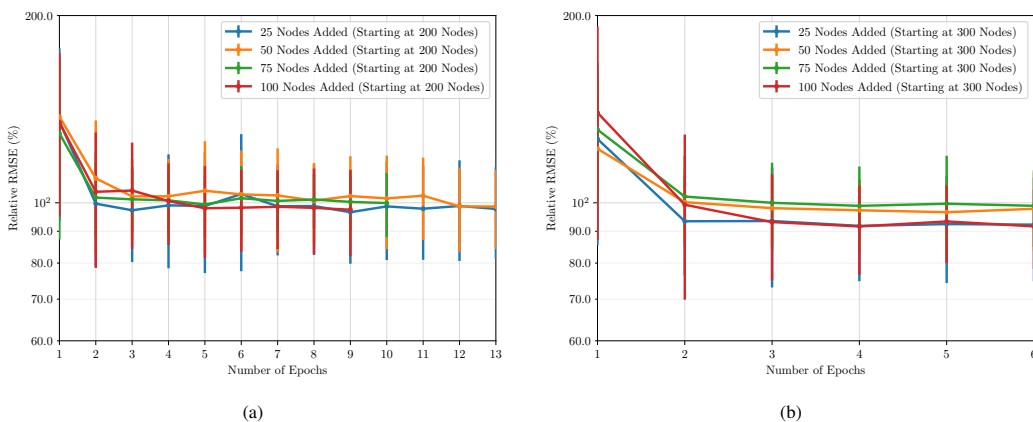

(a)  (b)

Figure 1: Relative RMSE of a recommendation system trained on MovieLens 100k for 20 independent partitions measured over the test set (a) starting with GNNs of 200 nodes, compared to a GNN trained on 1000 nodes for 37 epochs (b) starting with GNNs of 300 nodes, compared to a GNN trained on 1000 nodes for 29 epochs.

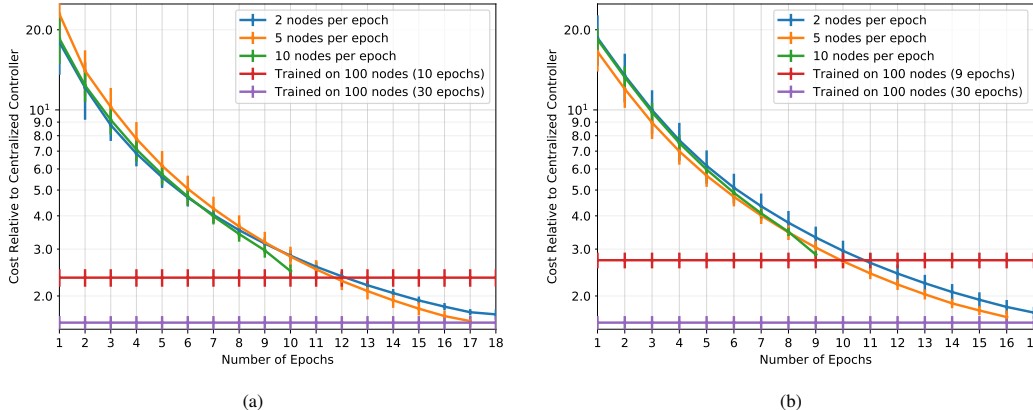

Figure 2: Velocity variation of the flocking problem for the whole trajectory in the testing set relative to the centralized controller (a) starting with 10 nodes and (b) starting with 20 nodes.

hand, agents are required to avoid collision. We can thus define the velocity variation of the team $\sigma_{\mathbf{v}(t)} = \sum_{i=1}^{n} \|v_i(t) - \bar{\mathbf{v}}(t)\|^2$, and the collision avoidance potential

$$CA(r_i, r_j) = \begin{cases} 1/\|r_i(t) - r_j(t)\|^2 - \log(\|r_i(t) - r_j(t)\|^2) & \text{if } \|r_i(t) - r_j(t)(t)\| \le R_{CA} \\ 1/R_{CA}^2 - \log(R_{CA}^2) & \text{otherwise,} \end{cases} \quad (18)$$

with $R_{CA} = 1m$. A centralized controller can be obtain by $u_i(t)^* = -n(v_i - \bar{\mathbf{v}}) + \sum_{j=1}^{n} \nabla_{r_i} CA(r_i, r_j)$ (Tanner et al., 2003).

Exploiting the fact that neural networks are universal approximators (Barron, 1993; Hornik, 1991), *Imitation Learning* can be utilized as a framework to train neural networks from information provided by an expert (Ross et al., 2011; Ross & Bagnell, 2010). Formally, we have a set of pairs $\mathcal{T} = \{\mathbf{x}_m, \mathbf{u}_m^*\}, m = 1, \ldots, M$, and during training we minimize the mean square error between the optimal centralized controller, and the output of our GNN $\|\mathbf{u}_m^* - \boldsymbol{\Phi}(\mathbf{x}_m; \mathcal{H}, \mathbf{S})\|^2$. Denoting $\mathcal{N}_i(t)$ the neighborhood of agent $i$ at time $t$, the state of the agents $\mathbf{x}(t) = [x(t)_1, \ldots, x(t)_n], x_i(t) \in \mathbb{R}^6$, is given by $x_i(t) = \sum_{j:j \in \mathcal{N}_i(t)} [v_i(t) - v_j(t), r_{ij}(t)/\|r_{ij}(t)\|^4, r_{ij}(t)/\|r_{ij}(t)\|^2]$. Note that state $x_i(t)$, gets transmitted between agents if a communication link exists between them.

In Figure 2 we can see the empirical manifestation of the claims we put forward. First and foremost, we are able to learn a GNN that achieves a comparable performance while taking steps on a smaller graphs. As seen in Figure 2, GNNs trained with $n_0 = \{10, 20\}$ agents in the first epoch and adding 10 agents per epoch (green line) are able to achieve a similar performance when reaching 100 agents than the one they would have achieved by training with 100 agents the same number of epochs. Besides, if we add less nodes per epoch, we are able to achieve a similar performance that we would have achieved by training on the large network for 30 epochs.

## 6 CONCLUSIONS

We have introduced a learning procedure for GNNs that progressively grows the size of the graph while training. Our algorithm requires less computational cost — as the number of nodes in the graph convolution is smaller — than training on the full graph without compromising performance. Leveraging transferability results, we bounded the expected difference between the gradient on the GNN, and the gradient on the WNN. Utilizing this result, we provided the theoretical guarantees that our Algorithm converges to a neighborhood of a first order stationary point of the WNN in finite time. We benchmarked our algorithm on a recommendation system and a decentralized control problem, achieving comparable performance to the one achieve by a GNN trained on the full graph.

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

## A  APPENDIX

## B  NUMERICAL RESULTS PARAMETERS

All experiments were done on a computer with 32gb of RAM, a CPU Intel Core i9-9900K @3.60GHz x 16, a GPU GeForce RTX 2080 Ti/PCIe/SSE2, and Ubuntu 18.04.4 LTS.

### B.1  RECOMMENDATION SYSTEM

We split the dataset with $90\%$ for the training set, and $10\%$ for the testing set, and we run 20 independent random partitions. For the optimizer, we used 5 samples for the batch size, and ADAM algorithm Kingma & Ba (2015), with learning rate 0.005, $\beta_1 = 0.9$, $\beta_2 = 0.999$, and without learning rate decay. For the loss, we used the smooth $L_1$ loss. For the GNN, we used ReLU as non-linearity, we considered $F = 32$ features, $K = 5$ filter taps, and $L = 1$ layers.

We used the graph neural networks library available online at https://github.com/alelab-upenn/graph-neural-networks/blob/master/examples/movieGNN.py and implemented with PyTorch.

### B.2  DECENTRALIZED CONTROL

We run the system for $T = 2s$, and used 400 samples for training, 20 for validation, and 20 for the test set. For the optimizer, we used 20 samples for the batch size, and ADAM algorithm Kingma & Ba (2015) with learning rate 0.0005, $\beta_1 = 0.9$, $\beta_2 = 0.999$, without learning rate decay. We used a one layer Graph Neural Networks with $F = 64$ hidden units and $K = 3$ filter taps, and used the hyperbolic tangent as non-linearity $\rho$. We run 10 independent realizations of each experiment.

We used the graph neural networks library available online at https://github.com/alelab-upenn/graph-neural-networks/blob/master/examples/flockingGNN.py and implemented with PyTorch.

## C  PROOF OF THEOREM 1

**Definition 2** (Template graphs). *Let $\{u_i\}_{i=1}^n$ be the regular $n$-partition of $[0,1]$, i.e.,*

$$u_i = \frac{i-1}{n} \tag{19}$$

*for $1 \le i \le n$. The $n$-node template graph $\mathbb{G}_n$, whose GSO we denote $\mathbb{S}_n$, is obtained from $\mathbf{W}$ as*

$$[\mathbb{S}_n]_{ij} = \mathbf{W}(u_i, u_j) \tag{20}$$

*for $1 \le i, j \le n$.*

**Definition 3** (Graphon spectral representation of convolutional filter response). *As the graphon $\mathbf{W}$ is bounded and symmetric, $T_{\mathbf{W}}$ is a self adjoint Hilbert-Schmidt operator, which allows to use the operator's spectral basis $\mathbf{W}(u,v) = \sum_{i\in\mathbf{Z}\{0\}} \lambda_i \psi_i(u)\psi_i(v)$. Eigenvalues $\lambda_i$ are ordered in decreasing order of absolute value i.e., $1 \ge \lambda_1 \ge \lambda_2 \ge \cdots \ge 0 \ge \cdots \ge \lambda_{-2} \ge \lambda_{-1} \ge -1$, and their only accumulation point is $0$ (Lax, 2002, Theorem 3, Chapter 28). Thus, we define the spectral representation of the convolutional filter $T_{\mathbf{H}}$ (cf. (6)) as,*

$$h(\lambda) = \sum_{k=0}^{K-1} h_k \lambda^k \tag{21}$$

**Definition 4** ($c$-band cardinality of $\mathbf{W}$). *The $c$-band cardinality, denoted $B_{\mathbf{W}}^c$, is the number of eigenvalues whose absolute value is larger than $c$.*

$$B_{\mathbf{W}}^c = \#\{\lambda_i : \|\lambda_i\| \le c\} \tag{22}$$

**Definition 5** ($c$-eigenvalue margin of $\mathbf{W}$ - $\mathbf{W}_n$). *The $c$-eigenvalue margin of $\mathbf{W}$ - $\mathbf{W}_n$ is defined as the minimum distance between two different eigenvalues of the integral operator applied to $\mathbf{W}$, and to $\mathbf{W}_n$ as follows,*

$$\delta_{\mathbf{W}\mathbf{W}_n}^c = \min_{i,j\ne i}\{\|\lambda_i(T_{\mathbf{W}}) - \lambda_i(T_{\mathbf{W}_n})\| : \|\lambda_i(T_{\mathbf{W}_n})\| \ge c\} \tag{23}$$

**Definition 6** (Graphon Convolutional Filter). *Given a graphon $\mathbf{W}$, a graphon signal $X$, and filter coefficients $\mathbf{h} = [h_0, \ldots, h_{K-1}]$ the graphon filter $T_{\mathbf{H}} : L_2([0,1]) \to L_2([0,1])$ is defined as,*

$$(T_{\mathbf{H}}X)(v) = \sum_{k=0}^{K-1} h_k (T_{\mathbf{W}}^{(k)} X)(v). \tag{24}$$

**Proposition 1.** *Let $X \in L_2([0,1])$ be a normalized Lipschitz graphon signal, and let $X_n$ be the graphon signal induced by the graph signal $\mathbf{x}_n$ obtained from $X$ on the template graph $\mathbb{G}_n$ [cf. Definition 2], i.e., $[\mathbf{x}_n]_i = X((i-1)/n)$ for $1 \le i \le n$. It holds that*

$$\|X - X_n\|_{L_2} \le \frac{1}{n}. \tag{25}$$

*Proof.* Let $I_i = [(i-1)/n, i/n)$ for $1 \le i \le n-1$ and $I_n = [(n-1)/n, 1]$. Since the graphon is normalized Lipschitz, for any $u \in I_i$, $1 \le i \le n$, we have

$$\|X(u) - X_n(u)\| \le \max\left(\left|u - \frac{i-1}{n}\right|, \left|\frac{i}{n} - u\right|\right) \le \frac{1}{n}. \tag{26}$$

We can then write

$$\|X - X_n\|^2 = \int_0^1 |X(u) - X_n(u)|^2 du \tag{27}$$

$$\le \int_0^1 \left(\frac{1}{n}\right)^2 du = \left(\frac{1}{n}\right)^2, \tag{28}$$

which completes the proof. $\qquad\square$

**Proposition 2.** *Let $\mathbf{W} : [0,1]^2 \to [0,1]$ be a normalized Lipschitz graphon, and let $\mathbb{W}_n := \mathbf{W}_{\mathbb{G}_n}$ be the graphon induced by the template graph $\mathbb{G}_n$ generated from $\mathbf{W}$ as in Definition 2. It holds that*

$$\|\mathbf{W} - \mathbb{W}_n\| \leq \frac{2}{n}. \tag{29}$$

*Proof.* Let $I_i = [(i-1)/n, i/n)$ for $1 \leq i \leq n-1$ and $I_n = [(n-1)/n, 1]$. Since the graphon is Lipschitz, for any $u \in I_i$, $v \in I_j$, $1 \leq i, j \leq n$, we have

$$\|\mathbf{W}(u,v) - \mathbb{W}_n(u,v)\| \leq \max\left(\left|u - \frac{i-1}{n}\right|, \left|\frac{i}{n} - u\right|\right) \tag{30}$$

$$+ \max\left(\left|v - \frac{j-1}{n}\right|, \left|\frac{j}{n} - v\right|\right) \tag{31}$$

$$\leq \frac{1}{n} + \frac{1}{n} = \frac{2}{n}. \tag{32}$$

We can then write

$$\|\mathbf{W} - \mathbb{W}_n\|^2 = \int_0^1 |\mathbf{W}(u,v) - \mathbb{W}_n(u,v)|^2 du dv \tag{33}$$

$$\leq \int_0^1 \left(\frac{2}{n}\right)^2 du dv = \left(\frac{2}{n}\right)^2 \tag{34}$$

which concludes the proof. $\square$

**Proposition 3.** *Consider the L-layer WNN given by $Y = \mathbf{\Phi}(X; \mathcal{H}, \mathbf{W})$, where $F_0 = F_L = 1$ and $F_\ell = F$ for $1 \leq \ell \leq L - 1$. Let $c \in (0, 1]$ and assume that the graphon convolutions in all layers of this WNN have $K$ filter taps [cf. (6)]. Under Assumptions 1 through 3, the norm of the gradient of the WNN with respect to its parameters $\mathcal{H} = \{\mathbf{H}_{lk}\}_{l,k}$ can be upper bounded by,*

$$\|\nabla_{\mathcal{H}} \mathbf{\Phi}(X; \mathcal{H}, \mathbf{W})\| \leq F^{2L} \sqrt{K}. \tag{35}$$

*Proof.* We will find an upper bound for any element $[\mathbf{H}_{l^\dagger k^\dagger}]_{g^\dagger f^\dagger}$ of the tensor $\mathcal{H}$. We start by the last layer of the WNN, applying the definition given in equation (8),

$$\|\nabla_{[\mathbf{H}_{l^\dagger k^\dagger}]_{g^\dagger f^\dagger}} \mathbf{\Phi}(X; \mathcal{H}, \mathbf{W})\| = \left\|\nabla_{[\mathbf{H}_{l^\dagger k^\dagger}]_{g^\dagger f^\dagger}} X_L^f\right\| \tag{36}$$

$$= \left\|\nabla_{[\mathbf{H}_{l^\dagger k^\dagger}]_{g^\dagger f^\dagger}} \rho\left(\sum_{g=1}^{F_{l-1}} \sum_{k=1}^{K-1} (T_{\mathbf{W}}^{(k)} X_{l-1}^g)[\mathbf{H}_{Lk}]_{gf}\right)\right\|. \tag{37}$$

By Assumption 3, the non-linearity $\rho$ is normalized Lipschitz , i.e. $\nabla\rho(\cdot)(u) \leq 1$ for all $u$. Thus, aplying the chain rule for the derivative, and the Cauchy-Schwartz inequality, the right hand side of the previous expression can be rewritten as,

$$\|\nabla_{[\mathbf{H}_{l^\dagger k^\dagger}]_{g^\dagger f^\dagger}} \mathbf{\Phi}(X; \mathcal{H}, \mathbf{W})\| = \left\|\nabla\rho\left(\sum_{g=1}^{F_{l-1}} \sum_{k=1}^{K-1} (T_{\mathbf{W}}^{(k)} X_{l-1}^g)[\mathbf{H}_{Lk}]_{gf}\right)\right\|$$

$$\left\|\nabla_{[\mathbf{H}_{l^\dagger k^\dagger}]_{g^\dagger f^\dagger}} \sum_{g=1}^{F_{l-1}} \sum_{k=1}^{K-1} (T_{\mathbf{W}}^{(k)} X_{l-1}^g)[\mathbf{H}_{Lk}]_{gf}\right\| \tag{38}$$

$$\leq \left\|\nabla_{[\mathbf{H}_{l^\dagger k^\dagger}]_{g^\dagger f^\dagger}} \sum_{g=1}^{F_{l-1}} \sum_{k=1}^{K-1} (T_{\mathbf{W}}^{(k)} X_{l-1}^g)[\mathbf{H}_{Lk}]_{gf}\right\| \tag{39}$$

Note that the a larger bound will occur if $l^\dagger < L - 1$, then by linearity of derivation, and the triangle inequality we obtain,

$$\|\nabla_{[\mathbf{H}_{l^\dagger k^\dagger}]_{g^\dagger f^\dagger}} \mathbf{\Phi}(X; \mathcal{H}, \mathbf{W})\| \leq \sum_{g=1}^{F_{l-1}} \left\|\sum_{k=1}^{K-1} T_{\mathbf{W}}^{(k)} (\nabla_{[\mathbf{H}_{l^\dagger k^\dagger}]_{g^\dagger f^\dagger}} X_{l-1}^g)[\mathbf{H}_{Lk}]_{gf}\right\| \tag{40}$$

By Assumption 2, the convolutional filters are non-amplifying, thus it holds that,

$$\|\nabla_{[\mathbf{H}_{l^\dagger k^\dagger}]_{g^\dagger f^\dagger}} \mathbf{\Phi}(X; \mathcal{H}, \mathbf{W})\| \leq \sum_{g=1}^{F_{l-1}} \left\| \nabla_{[\mathbf{H}_{l^\dagger k^\dagger}]_{g^\dagger f^\dagger}} X_{l-1}^g \right\| \tag{41}$$

Now note that as filters are non-amplifying, the maximum difference in the gradient will be attained at the first layer ($l = 1$) of the WNN. Also note that the derivative of a convolutional filter $T_{\mathbf{H}}$ [cf. Definition 6] at coefficient $k^\dagger = i$, is itself a convolutional filter with coefficients $\mathbf{h}_i$. The values of $\mathbf{h}_i$ are $[\mathbf{h}_i]_j = 1$ if $j = i$ and 0 otherwise. Thence,

$$\|\nabla_{[\mathbf{H}_{l^\dagger k^\dagger}]_{g^\dagger f^\dagger}} \mathbf{\Phi}(X; \mathcal{H}, \mathbf{W})\| \leq F^{L-1} \left\| \mathbf{h}_{i*\mathbf{W}} X_0 \right\| \tag{42}$$

$$\leq F^{L-1} \|X_0\|. \tag{43}$$

To complete the proof note that tensor $\mathcal{H}$ has $F^{L-1}K$ elements, and each individual gradient is upper bounded by (43), and $\|X\|$ is normalized by Assumption 1. $\square$

**Lemma 1.** *Let* $\mathbf{\Phi}(X; \mathcal{H}, \mathbf{W})$ *be a WNN with* $F_0 = F_L = 1$, *and* $F_l = F$ *for* $1 \leq l \leq L-1$. *Let* $c \in (0, 1]$, *and assume that the graphon convolutions in all layers of this WNN have* $K$ *filter taps [cf. (6)]. Let* $\mathbf{\Phi}(\mathbf{x}_n; \mathcal{H}, \mathbf{S}_n)$ *be a GNN sampled from* $\mathbf{\Phi}(X; \mathcal{H}, \mathbf{W})$ *as in (9). Under assumptions* (1),(2),(3), *and* (5) *with probability* $1 - \xi$ *it holds that,*

$$\|\mathbf{\Phi}(X; \mathcal{H}, \mathbf{W}) - \mathbf{\Phi}(X_n; \mathcal{H}, \mathbf{W}_n)\| \leq LF^{L-1} \left(1 + \frac{\pi B_{\mathbf{W}_n}^c}{\delta_{\mathbf{W}\mathbf{W}_n}^c}\right) \frac{2\left(1 + \sqrt{n \log(\frac{2n}{\xi})}\right)}{n}$$
$$+ \frac{1}{n} + 4LF^{L-1}c \tag{44}$$

*The fixed constants* $B_{\mathbf{W}}^c$ *and* $\delta_{\mathbf{W}\mathbf{W}_n}^c$ *are the c-band cardinality and the c-eigenvalue margin of* $\mathbf{W}$ *and* $\mathbf{W}_n$ *respectively [cf. Definitions 4,5].*

*Proof.* We start by writing the expression on the left hand side, using the definition of WNN [cf. (8)] we can write,

$$\|\mathbf{\Phi}(X; \mathcal{H}, \mathbf{W}) - \mathbf{\Phi}(X_n; \mathcal{H}, \mathbf{W}_n)\| = \|X_L - X_{nL}\| \tag{45}$$

$$= \left\| \rho\left(\sum_{g=1}^{F_{L-1}} \sum_{k=1}^{K-1} (T_{\mathbf{W}}^{(k)} X_{L-1}^g)[\mathbf{H}_{Lk}]_{gf}\right) - \rho\left(\sum_{g=1}^{F_{L-1}} \sum_{k=1}^{K-1} (T_{\mathbf{W}_n}^{(k)} X_{nL-1}^g)[\mathbf{H}_{Lk}]_{gf}\right) \right\|.$$

Since the non-linearity $\rho$ is normalized Lipschitz by Assumption 3, using the triangle inequality, we obtain

$$\|X_L - X_{nL}\| \leq \sum_{g=1}^{F_{L-1}} \left\| \sum_{k=1}^{K-1} (T_{\mathbf{W}}^{(k)} X_{L-1}^g)[\mathbf{H}_{Lk}]_{gf} - \sum_{k=1}^{K-1} (T_{\mathbf{W}_n}^{(k)} X_{nL-1}^g)[\mathbf{H}_{Lk}]_{gf} \right\|. \tag{46}$$

Using the triangle inequality once again, we split the last inequality into two terms as follows,

$$\|X_L - X_{nL}\| \leq \sum_{g=1}^{F_{L-1}} \left\| \sum_{k=1}^{K-1} T_{\mathbf{W}}^{(k)} (X_{L-1}^g - X_{nL-1}^g)[\mathbf{H}_{Lk}]_{gf} \right\| \quad \mathbf{(1)}$$

$$+ \sum_{g=1}^{F_{L-1}} \left\| \sum_{k=1}^{K-1} (T_{\mathbf{W}}^{(k)} - T_{\mathbf{W}_n}^{(k)}) X_{L-1}^g [\mathbf{H}_{Lk}]_{gf} \right\| \quad \mathbf{(2)}. \tag{47}$$

Where we have split (47) into terms **(1)**, and **(2)**. On the one hand, by assumption 2, convolutional filters $h$ are non-amplifying, thus using Cauchy-Schwartz inequality, term **(1)** can be bounded by,

$$\sum_{g=1}^{F_{L-1}} \left\| \sum_{k=1}^{K-1} T_{\mathbf{W}}^{(k)} (X_{L-1}^g - X_{nL-1}^g)[\mathbf{H}_{Lk}]_{gf} \right\| \leq \sum_{g=1}^{F_{L-1}} \left\| X_{L-1}^g - X_{nL-1}^g \right\|. \tag{48}$$

To bound term **(2)**, denoting $h_{Lgf}$ the spectral representation of the convolutional filter applied to $X_{L-1}^g$ at feature $f$ of layer $L$ [cf. Definition 3], we will decompose the filter as follows,

$$h_{Lgf}^{\geq c}(\lambda) \begin{cases} 0 & \text{if } |\lambda| < c \\ h_{Lgf}(\lambda) - h_{Lgf}(c) & \text{if } |\lambda| \geq c \end{cases} \tag{49}$$

$$h_{Lgf}^{<c}(\lambda) \begin{cases} h_{Lgf}(\lambda) & \text{if } |\lambda| < c \\ h_{Lgf}(c) & \text{if } |\lambda| \geq c. \end{cases} \tag{50}$$

Note that $h_{Lgf} = h_{Lgf}^{\geq c} + h_{Lgf}^{<c}$. Let $T_{[\mathbf{H}_L]_{gf}}^{<c}$ and $T_{[\mathbf{H}_{nL}]_{gf}}^{<c}$, be the graphon convolutional filters with filter function $h_{Lgf}^{<c}$ on graphons $\mathbf{W}$, and $\mathbf{W}_n$ respectively [cf. Definition 6]. Note that filter $h_{Lgf}^{<c}$, varies only in the interval $[0, c)$, and since filters are normalized Lipschitz by Assumption 2, it verifies

$$\left\| T_{[\mathbf{H}_L]_{gf}}^{<c} X_{L-1}^g - T_{[\mathbf{H}_{nL}]_{gf}}^{<c} X_{L-1}^g \right\| \leq \left\| (h_{Lgf}(c) + c) - (h_{Lgf}(c) - c) \| \| X_{L-1}^g \right\| \tag{51}$$

$$\leq 2c \| X_{L-1}^g \|. \tag{52}$$

Now we need to upper bound the difference in the high frequencies $h_{Lgf}^{\geq c}$. Let $T_{[\mathbf{H}_L]_{gf}}^{\geq c}$ and $T_{[\mathbf{H}_{nL}]_{gf}}^{\geq c}$, be the graphon filters with filter function $h_{Lgf}^{\geq c}$ on graphons $\mathbf{W}$, and $\mathbf{W}_n$ respectively. Let $\mathbb{S}_n$ denote the template graph sampled from the graphon $\mathbf{W}$ [cf. definition 2]. We denote $\mathbb{W}_n$, the induced graphon by template graph $\mathbb{S}_n$ as in (10). By introducing $T_{[\mathbb{H}_{nL}]_{gf}}^{\geq c}$, the graph filter with filter function $h_{Lgf}^{\geq c}$ on graphon $\mathbb{W}_n$, we can use the triangle inequality to obtain,

$$\left\| T_{[\mathbf{H}_L]_{gf}}^{\geq c} X_{L-1}^g - T_{[\mathbf{H}_{nL}]_{gf}}^{\geq c} X_{L-1}^g \right\| \leq \left\| T_{[\mathbf{H}_L]_{gf}}^{\geq c} X_{L-1}^g - T_{[\mathbb{H}_{nL}]_{gf}}^{\geq c} X_{L-1}^g \right\| \textbf{(2.1)}$$

$$+ \left\| T_{[\mathbb{H}_{nL}]_{gf}}^{\geq c} X_{L-1}^g - T_{[\mathbf{H}_{nL}]_{gf}}^{\geq c} X_{L-1}^g \right\| \textbf{(2.2)}. \tag{53}$$

Under assumptions 1–5, to bound term **(2.1)** we can use (Ruiz et al., 2020a, Theorem 1), and to bound term **(2.2)** we can use (Ruiz et al., 2020d, Lemma 2). Thus, with probability $1 - \xi$, the previous expression can be bounded by,

$$\left\| T_{[\mathbf{H}_L]_{gf}}^{\geq c} X_{L-1}^g - T_{[\mathbf{H}_{nL}]_{gf}}^{\geq c} X_{L-1}^g \right\| \leq \left( 1 + \frac{\pi B_{\mathbf{W}_n}^c}{\delta_{\mathbf{W}\mathbf{W}_n}^c} \right) \frac{2 \left( 1 + \sqrt{n \log(\frac{2n}{\xi})} \right)}{n} \left\| X_{L-1}^g \right\|. \tag{54}$$

Where the fixed constants $B_{\mathbf{W}}^c$ and $\delta_{\mathbf{W}\mathbf{W}_n}^c$ are the $c$-band cardinality and the $c$-eigenvalue margin of $\mathbf{W}$ and $\mathbf{W}_n$ respectively [cf. Definitions 4,5]. Hence, coming back to (47), we can use (48) to upper bound **(1)**, and we can use (52), and (54), to upper bound **(2)** as follows,

$$\| X_L - X_{nL} \| \leq \sum_{g=1}^{F_{L-1}} \left\| X_{L-1}^g - X_{nL-1}^g \right\| + 2c \| X_{L-1}^g \|$$

$$+ \left( 1 + \frac{\pi B_{\mathbf{W}_n}^c}{\delta_{\mathbf{W}\mathbf{W}_n}^c} \right) \frac{2 \left( 1 + \sqrt{n \log(\frac{2n}{\xi})} \right)}{n} \left\| X_{L-1}^g \right\|. \tag{55}$$

Now, we arrive at a recursive equation that we can compute for the $L$ layers, with $F$ features per layer, to obtain,

$$\| X_L - X_{nL} \| \leq F_0 \| X_0 - X_{n0} \| + 2LF^{L-1} c \| X_0 \|$$

$$+ LF^{L-1} \left( 1 + \frac{\pi B_{\mathbf{W}_n}^c}{\delta_{\mathbf{W}\mathbf{W}_n}^c} \right) \frac{2 \left( 1 + \sqrt{n \log(\frac{2n}{\xi})} \right)}{n} \| X_0 \|. \tag{56}$$

Using Proposition 1, noting that $F_0 = 1$ by construction, and using Assumption 1, concludes the proof. $\qquad\square$

**Lemma 2.** *Let $\Phi(X; \mathcal{H}, \mathbf{W})$ be a WNN with $F_0 = F_L = 1$, and $F_l = F$ for $1 \leq l \leq L - 1$. Let $c \in (0, 1]$, and assume that the graphon convolutions in all layers of this WNN have $K$ filter taps [cf. (6)]. Let $\Phi(\mathbf{x}_n; \mathcal{H}, \mathbf{S}_n)$ be a GNN sampled from $\Phi(X; \mathcal{H}, \mathbf{W})$ as in (9). Under assumptions (1),(2),(3), and (5) with probability $1 - \xi$ it holds that,*

$$\|\nabla_{\mathcal{H}} \Phi(X; \mathcal{H}, \mathbf{W}) - \nabla_{\mathcal{H}} \Phi(X_n; \mathcal{H}, \mathbf{W}_n)\|$$

$$\leq \sqrt{KF^{L-1}} \left( 2L^2 F^{2L-2} \left( 1 + \frac{\pi B_{\mathbf{W}_n}^c}{\delta_{\mathbf{W}\mathbf{W}_n}^c} \right) \frac{2 \left( 1 + \sqrt{n \log(\frac{2n}{\xi})} \right)}{n} \right.$$

$$\left. + \frac{2F^{L-1}L}{n} + 8L^2 F^{2L-2} c \right).$$

*Proof.* We will first show that the gradient with respect to any arbitrary element $[\mathbf{H}_{l^\dagger k^\dagger}]_{g^\dagger f^\dagger} \in \mathbb{R}$ of $\mathcal{H}$ can be uniformly bounded. Note that the maximum is attained if $l^\dagger = 1$. Without loss of generality, assuming $l^\dagger > l - 1$, we can begin by using the definition given in equation (8) of the output of the WNN as follows,

$$\|\nabla_{[\mathbf{H}_{l^\dagger k^\dagger}]_{g^\dagger f^\dagger}} \Phi(X; \mathcal{H}, \mathbf{W}) - \nabla_{[\mathbf{H}_{l^\dagger k^\dagger}]_{g^\dagger f^\dagger}} \Phi(X_n; \mathcal{H}, \mathbf{W}_n)\|$$

$$= \|\nabla_{[\mathbf{H}_{l^\dagger k^\dagger}]_{g^\dagger f^\dagger}} X_L^f - \nabla_{[\mathbf{H}_{l^\dagger k^\dagger}]_{g^\dagger f^\dagger}} X_{nL}^f\| \tag{57}$$

$$= \left\| \nabla_{[\mathbf{H}_{l^\dagger k^\dagger}]_{g^\dagger f^\dagger}} \rho \left( \sum_{g=1}^{F_{l-1}} \sum_{k=1}^{K-1} (T_{\mathbf{W}}^{(k)} X_{l-1}^g)[\mathbf{H}_{lk}]_{gf} \right) \right.$$

$$\left. - \nabla_{[\mathbf{H}_{l^\dagger k^\dagger}]_{g^\dagger f^\dagger}} \rho \left( \sum_{g=1}^{F_{l-1}} \sum_{k=1}^{K-1} (T_{\mathbf{W}_n}^{(k)} X_{nl-1}^g)[\mathbf{H}_{lk}]_{gf} \right) \right\|. \tag{58}$$

Taking derivatives by applying the chain rule, and applying the triangle inequality it yields,

$$\|\nabla_{[\mathbf{H}_{l^\dagger k^\dagger}]_{g^\dagger f^\dagger}} X_L^f - \nabla_{[\mathbf{H}_{l^\dagger k^\dagger}]_{g^\dagger f^\dagger}} X_{nL}^f\|$$

$$\leq \left\| \left( \nabla \rho \left( \sum_{g=1}^{F_{l-1}} \sum_{k=1}^{K-1} (T_{\mathbf{W}}^{(k)} X_{l-1}^g)[\mathbf{H}_{lk}]_{gf} \right) - \nabla \rho \left( \sum_{g=1}^{F_{l-1}} \sum_{k=1}^{K-1} (T_{\mathbf{W}_n}^{(k)} X_{nl-1}^g)[\mathbf{H}_{lk}]_{gf} \right) \right) \right.$$

$$\nabla_{[\mathbf{H}_{l^\dagger k^\dagger}]_{g^\dagger f^\dagger}} \left( \sum_{g=1}^{F_{l-1}} \sum_{k=1}^{K-1} (T_{\mathbf{W}}^{(k)} X_{l-1}^g)[\mathbf{H}_{lk}]_{gf} \right) \right\| \tag{59}$$

$$+ \left\| \nabla \rho \left( \sum_{g=1}^{F_{l-1}} \sum_{k=1}^{K-1} (T_{\mathbf{W}_n}^{(k)} X_{nl-1}^g)[\mathbf{H}_{lk}]_{gf} \right) \right. \tag{60}$$

$$\left. \left( \nabla_{[\mathbf{H}_{l^\dagger k^\dagger}]_{g^\dagger f^\dagger}} \sum_{g=1}^{F_{l-1}} \sum_{k=1}^{K-1} (T_{\mathbf{W}}^{(k)} X_{l-1}^g)[\mathbf{H}_{lk}]_{gf} - \nabla_{[\mathbf{H}_{l^\dagger k^\dagger}]_{g^\dagger f^\dagger}} \sum_{g=1}^{F_{l-1}} \sum_{k=1}^{K-1} (T_{\mathbf{W}_n}^{(k)} X_{nl-1}^g)[\mathbf{H}_{lk}]_{gf} \right) \right\|.$$

We can now use Cauchy-Schwartz inequality, Assumptions 3, 4, and Proposition 3 to bound the terms regarding the gradient of the non-linearity $\rho$, the loss function $\ell$, and the WNN respectively, as follows,

$$\|\nabla_{[\mathbf{H}_{l^\dagger k^\dagger}]_{g^\dagger f^\dagger}} X_L^f - \nabla_{[\mathbf{H}_{l^\dagger k^\dagger}]_{g^\dagger f^\dagger}} X_{nL}^f\| \tag{61}$$

$$\leq \left\| \sum_{g=1}^{F_{l-1}} \sum_{k=1}^{K-1} (T_{\mathbf{W}}^{(k)} X_{l-1}^g)[\mathbf{H}_{lk}]_{gf} - \sum_{g=1}^{F_{l-1}} \sum_{k=1}^{K-1} (T_{\mathbf{W}_n}^{(k)} X_{nl-1}^g)[\mathbf{H}_{lk}]_{gf} \right\| F^{L-1} \|X_0\|$$

$$+ \left\| \sum_{g=1}^{F_{l-1}} \nabla_{[\mathbf{H}_{l^\dagger k^\dagger}]_{g^\dagger f^\dagger}} \sum_{k=1}^{K-1} \left( (T_{\mathbf{W}}^{(k)} X_{l-1}^g)[\mathbf{H}_{lk}]_{gf} - (T_{\mathbf{W}_n}^{(k)} X_{nl-1}^g)[\mathbf{H}_{lk}]_{gf} \right) \right\|.$$

We can now apply the triangle inequality on the second term of the previous bound to obtain,

$$\|\nabla_{[\mathbf{H}_{l^\dagger k^\dagger}]_{g^\dagger f^\dagger}} X_L^f - \nabla_{[\mathbf{H}_{l^\dagger k^\dagger}]_{g^\dagger f^\dagger}} X_{nL}^f\| \tag{62}$$

$$\leq \left\| \sum_{g=1}^{F_{l-1}} \sum_{k=1}^{K-1} (T_{\mathbf{W}}^{(k)} X_{l-1}^g)[\mathbf{H}_{lk}]_{gf} - \sum_{g=1}^{F_{l-1}} \sum_{k=1}^{K-1} (T_{\mathbf{W}_n}^{(k)} X_{nl-1}^g)[\mathbf{H}_{lk}]_{gf} \right\| F^{L-1} \|X_0\|$$

$$+ \left\| \sum_{g=1}^{F_{l-1}} \nabla_{[\mathbf{H}_{l^\dagger k^\dagger}]_{g^\dagger f^\dagger}} \sum_{k=1}^{K-1} \left( (T_{\mathbf{W}}^{(k)})[\mathbf{H}_{lk}]_{gf} - (T_{\mathbf{W}_n}^{(k)})[\mathbf{H}_{lk}]_{gf} \right) X_{nl-1}^g \right\|$$

$$+ \sum_{g=1}^{F_{l-1}} \left\| \nabla_{[\mathbf{H}_{l^\dagger k^\dagger}]_{g^\dagger f^\dagger}} \sum_{k=1}^{K-1} T_{\mathbf{W}_n}^{(k)} \left( X_{l-1}^g - X_{nl-1}^g \right) [\mathbf{H}_{lk}]_{gf}) \right\|.$$

Now note that as we are considering the case in which $l_\dagger < l-1$, using Cauchy-Schwartz inequality, we can use the same bound for the first and second term of the right hand side of the previous inequality. Since filters are non-expansive by Assumption 3, it yields

$$\|\nabla_{[\mathbf{H}_{l^\dagger k^\dagger}]_{g^\dagger f^\dagger}} X_L^f - \nabla_{[\mathbf{H}_{l^\dagger k^\dagger}]_{g^\dagger f^\dagger}} X_{nL}^f\| \tag{63}$$

$$\leq 2 \left\| \sum_{g=1}^{F_{l-1}} \sum_{k=1}^{K-1} (T_{\mathbf{W}}^{(k)} X_{l-1}^g)[\mathbf{H}_{lk}]_{gf} - \sum_{g=1}^{F_{l-1}} \sum_{k=1}^{K-1} (T_{\mathbf{W}_n}^{(k)} X_{nl-1}^g)[\mathbf{H}_{lk}]_{gf} \right\| F^{L-1} \|X_0\|$$

$$+ \sum_{g=1}^{F_{l-1}} \left\| \nabla_{[\mathbf{H}_{l^\dagger k^\dagger}]_{g^\dagger f^\dagger}} \left( X_{l-1}^g - X_{nl-1}^g \right) \right\|.$$

Now notice, that the only term that remains to bound is the exact same bound we obtained in equation (57), but on the previous layer $L-2$. Hence, we conclude that by applying the same steps $L-2$ times, as the WNN has $L$ layers, we will obtain a bound for any element $[\mathbf{H}_{l^\dagger k^\dagger}]_{g^\dagger f^\dagger}$ of tensor $\mathcal{H}$.

$$\|\nabla_{[\mathbf{H}_{l^\dagger k^\dagger}]_{g^\dagger f^\dagger}} X_L^f - \nabla_{[\mathbf{H}_{l^\dagger k^\dagger}]_{g^\dagger f^\dagger}} X_{nL}^f\| \tag{64}$$

$$\leq 2LF^{L-2} \left\| \sum_{g=1}^{F_{l-1}} \sum_{k=1}^{K-1} (T_{\mathbf{W}}^{(k)} X_{l-1}^g)[\mathbf{H}_{lk}]_{gf} - \sum_{g=1}^{F_{l-1}} \sum_{k=1}^{K-1} (T_{\mathbf{W}_n}^{(k)} X_{nl-1}^g)[\mathbf{H}_{lk}]_{gf} \right\| F^{L-1} \|X_0\|$$

$$+ \sum_{g=1}^{F_{l-1}} \left\| \nabla_{[\mathbf{H}_{l^\dagger k^\dagger}]_{g^\dagger f^\dagger}} \left( X_1^g - X_1^g \right) \right\|.$$

Note that the derivative of a convolutional filter $T_{\mathbf{H}}$ at coefficient $k^\dagger = i$, is itself a convolutional filter with coefficients $\mathbf{h}_i$ [cf. Definition 6]. The values of $\mathbf{h}_i$ are $[\mathbf{h}_i]_j = 1$ if $j = i$ and 0 otherwise. As $\mathbf{h}_i$ is itself a filter that verifies Assumption 2, as graphons are normalized. Thus, considering $l^\dagger = 0$, and using Propositions 1, 2, (Chung & Radcliffe, 2011, Theorem 1) and the triangle inequality, we obtain,

$$\left\| \mathbf{h}_{i*\mathbf{W}_n} X_{n0} - \mathbf{h}_{i*\mathbf{W}} X_0 \right\| \leq \left( \|\mathbf{W} - \mathbb{W}_n\| + \|\mathbb{W}_n - \mathbf{W}_n\| \right) \|X_0\| + \|X_{n0} - X_0\| \tag{65}$$

$$\leq \left( 1 + \frac{\pi B_{\mathbf{W}_n}^c}{\delta_{\mathbf{W}\mathbf{W}_n}^c} \right) \frac{2\left(1 + \sqrt{n \log(\frac{2n}{\xi})}\right)}{n} + \frac{1}{n} \tag{66}$$

with probability $1 - \xi$. In the previous expression, $\mathbb{W}_n$ is the template graphon [cf. Definition 2]. Now, substituting (64) into (65), and using Lemma 1, with probability $1 - \xi$, it holds that,

$$\|\nabla_{[\mathbf{H}_{l^\dagger k^\dagger}]_{g^\dagger f^\dagger}} X_L^f - \nabla_{[\mathbf{H}_{l^\dagger k^\dagger}]_{g^\dagger f^\dagger}} X_{nL}^f\| \leq 2L^2 F^{2L-2} \left( 1 + \frac{\pi B_{\mathbf{W}_n}^c}{\delta_{\mathbf{W}\mathbf{W}_n}^c} \right) \frac{2\left(1 + \sqrt{n \log(\frac{2n}{\xi})}\right)}{n}$$

$$+ \frac{2F^{L-1}L}{n} + 8L^2 F^{2L-2} c. \tag{67}$$

To achieve the final result, note that tensor $\mathcal{H}$ has $KF^{L-1}$ elements, and each individual gradient is upper bounded by (67). ☐

**Lemma 3.** *Let $\mathbf{\Phi}(X; \mathcal{H}, \mathbf{W})$ be a WNN with $F_0 = F_L = 1$, and $F_l = F$ for $1 \leq l \leq L - 1$. Let $c \in (0, 1]$, and assume that the graphon convolutions in all layers of this WNN have $K$ filter taps [cf. (6)]. Let $\mathbf{\Phi}(\mathbf{x}_n; \mathcal{H}, \mathbf{S}_n)$ be a GNN sampled from $\mathbf{\Phi}(X; \mathcal{H}, \mathbf{W})$ as in (9). Under Assumptions (1)–(5) with probability $1 - \xi$ it holds that,*

$$\|\nabla_{\mathcal{H}} \ell(Y, \mathbf{\Phi}(X; \mathcal{H}, \mathbf{W})) - \nabla_{\mathcal{H}} \ell(Y_n, \mathbf{\Phi}(X_n; \mathcal{H}, \mathbf{W}_n))\|$$

$$\leq \sqrt{KF^{L-1}} \left( 3L^2 F^{2L-2} \left( 1 + \frac{\pi B_{\mathbf{W}_n}^c}{\delta_{\mathbf{W}\mathbf{W}_n}^c} \right) \frac{2\left( 1 + \sqrt{n \log(\frac{2n}{\xi})} \right)}{n} \right.$$

$$\left. + \frac{4F^{L-1}L}{n} + 12L^2 F^{2L-2} c \right)$$

*Proof.* In order to analyze the norm of the gradient with respect to the tensor $\mathcal{H}$, we can start by taking the derivative with respect to a single element of the tensor, $[\mathbf{H}_{l\dagger k\dagger}]_{g\dagger f\dagger}$. By deriving the loss function $\ell$ using the chain rule it yields,

$$\|\nabla_{[\mathbf{H}_{l\dagger k\dagger}]_{g\dagger f\dagger}} \ell(Y, \mathbf{\Phi}(X; \mathcal{H}, \mathbf{W})) - \nabla_{[\mathbf{H}_{l\dagger k\dagger}]_{g\dagger f\dagger}} \ell(Y_n, \mathbf{\Phi}(X_n; \mathcal{H}, \mathbf{W}_n))\|$$

$$= \|\nabla \ell(Y, \mathbf{\Phi}(X; \mathcal{H}, \mathbf{W})) \nabla_{[\mathbf{H}_{l\dagger k\dagger}]_{g\dagger f\dagger}} \mathbf{\Phi}(X; \mathcal{H}, \mathbf{W})$$

$$- \nabla \ell(Y_n, \mathbf{\Phi}(X_n; \mathcal{H}, \mathbf{W}_n)) \nabla_{[\mathbf{H}_{l\dagger k\dagger}]_{g\dagger f\dagger}} \mathbf{\Phi}(X_n; \mathcal{H}, \mathbf{W}_n)\|. \tag{68}$$

By Cauchy-Schwartz, and the triangle inequality it holds,

$$\|\nabla_{[\mathbf{H}_{l\dagger k\dagger}]_{g\dagger f\dagger}} \ell(Y, \mathbf{\Phi}(X; \mathcal{H}, \mathbf{W})) - \nabla_{[\mathbf{H}_{l\dagger k\dagger}]_{g\dagger f\dagger}} \ell(Y_n, \mathbf{\Phi}(X_n; \mathcal{H}, \mathbf{W}_n))\|$$

$$\leq \|\nabla \ell(Y, \mathbf{\Phi}(X; \mathcal{H}, \mathbf{W})) - \nabla \ell(Y_n, \mathbf{\Phi}(X_n; \mathcal{H}, \mathbf{W}_n))\| \|\nabla_{[\mathbf{H}_{l\dagger k\dagger}]_{g\dagger f\dagger}} \mathbf{\Phi}(X; \mathcal{H}, \mathbf{W})\| \tag{69}$$

$$+ \|\nabla \ell(Y_n, \mathbf{\Phi}(X_n; \mathcal{H}, \mathbf{W}_n))\| \|\nabla_{[\mathbf{H}_{l\dagger k\dagger}]_{g\dagger f\dagger}} \mathbf{\Phi}(X; \mathcal{H}, \mathbf{W}) - \nabla_{[\mathbf{H}_{l\dagger k\dagger}]_{g\dagger f\dagger}} \mathbf{\Phi}(X_n; \mathcal{H}, \mathbf{W}_n)\|.$$

By the triangle inequality and Assumption 4 it follows,

$$\|\nabla_{[\mathbf{H}_{l\dagger k\dagger}]_{g\dagger f\dagger}} \ell(Y, \mathbf{\Phi}(X; \mathcal{H}, \mathbf{W})) - \nabla_{[\mathbf{H}_{l\dagger k\dagger}]_{g\dagger f\dagger}} \ell(Y_n, \mathbf{\Phi}(X_n; \mathcal{H}, \mathbf{W}_n))\|$$

$$\leq \|\nabla \ell(Y, \mathbf{\Phi}(X; \mathcal{H}, \mathbf{W})) - \nabla \ell(Y, \mathbf{\Phi}(X_n; \mathcal{H}, \mathbf{W}_n))\| \|\nabla_{[\mathbf{H}_{l\dagger k\dagger}]_{g\dagger f\dagger}} \mathbf{\Phi}(X; \mathcal{H}, \mathbf{W})\| \tag{70}$$

$$\|\nabla \ell(Y_n, \mathbf{\Phi}(X_n; \mathcal{H}, \mathbf{W}_n)) - \nabla \ell(Y, \mathbf{\Phi}(X_n; \mathcal{H}, \mathbf{W}_n))\| \|\nabla_{[\mathbf{H}_{l\dagger k\dagger}]_{g\dagger f\dagger}} \mathbf{\Phi}(X; \mathcal{H}, \mathbf{W})\| \tag{71}$$

$$+ \|\nabla_{[\mathbf{H}_{l\dagger k\dagger}]_{g\dagger f\dagger}} \mathbf{\Phi}(X; \mathcal{H}, \mathbf{W}) - \nabla_{[\mathbf{H}_{l\dagger k\dagger}]_{g\dagger f\dagger}} \mathbf{\Phi}(X_n; \mathcal{H}, \mathbf{W}_n)\|$$

$$\leq (\|Y_n - Y\| + \|\mathbf{\Phi}(X_n; \mathcal{H}, \mathbf{W}_n)) - \mathbf{\Phi}(X; \mathcal{H}, \mathbf{W}))\|) \|\nabla_{[\mathbf{H}_{l\dagger k\dagger}]_{g\dagger f\dagger}} \mathbf{\Phi}(X; \mathcal{H}, \mathbf{W})\| \tag{72}$$

$$+ \|\nabla_{[\mathbf{H}_{l\dagger k\dagger}]_{g\dagger f\dagger}} \mathbf{\Phi}(X; \mathcal{H}, \mathbf{W}) - \nabla_{[\mathbf{H}_{l\dagger k\dagger}]_{g\dagger f\dagger}} \mathbf{\Phi}(X_n; \mathcal{H}, \mathbf{W}_n)\|.$$

Now we can use Lemmas 1–2, Propositions 1, and 3, and Assumption 1 to obtain,

$$\|\nabla_{[\mathbf{H}_{l\dagger k\dagger}]_{g\dagger f\dagger}} \ell(Y, \mathbf{\Phi}(X; \mathcal{H}, \mathbf{W})) - \nabla_{[\mathbf{H}_{l\dagger k\dagger}]_{g\dagger f\dagger}} \ell(Y_n, \mathbf{\Phi}(X_n; \mathcal{H}, \mathbf{W}_n))\| \tag{73}$$

$$\leq \left( 3L^2 F^{2L-2} \left( 1 + \frac{\pi B_{\mathbf{W}_n}^c}{\delta_{\mathbf{W}\mathbf{W}_n}^c} \right) \frac{2\left( 1 + \sqrt{n \log(\frac{2n}{\xi})} \right)}{n} \right.$$

$$\left. + \frac{4F^{L-1}L}{n} + 12L^2 F^{2L-2} c \right)$$

Noting that tensor $\mathcal{H}$ has $KF^{L-1}$ elements, and each individual term can be bounded by (73), the desired result is attained. $\qquad \square$

**Definition 7.** *We define the constant $\gamma$ as,*

$$\gamma = 12\sqrt{KF^{L-1}}L^2 F^{2L-2}, \tag{74}$$

*where $K$ is the number of features, $L$ is the number of layers, and $K$ is the number of filter taps of the GNN.*

We will present a more comprehensive statement of Theorem 1, where we include all the smaller order terms in (15). Notice that the statement of Theorem 1 in the main body of the paper omits these terms in order to simplify the exposition of the main result. In practice, these smaller order terms vanish faster as $n$ increases.

**Theorem 1.** *Consider the ERM problem in* (12) *and let* $\Phi(X; \mathcal{H}, \mathbf{W})$ *be an L-layer WNN with* $F_0 = F_L = 1$, *and* $F_l = F$ *for* $1 \le l \le L - 1$. *Let* $c \in (0, 1]$ *and assume that the graphon convolutions in all layers of this WNN have K filter taps [cf.* (6)*]. Let* $\Phi(\mathbf{x}_n; \mathcal{H}, \mathbf{S}_n)$ *be a GNN sampled from* $\Phi(X; \mathcal{H}, \mathbf{W})$ *as in* (9)*. Under assumptions AS1–AS5, it holds that*

$$\mathbb{E}[\|\nabla_{\mathcal{H}}\ell(Y, \Phi(X; \mathcal{H}, \mathbf{W})) - \nabla_{\mathcal{H}}\ell(Y_n, \Phi(X_n; \mathcal{H}, \mathbf{W}_n))\|]$$

$$\le \sqrt{KF^{L-1}}\left(6L^2F^{2L-2}\left(1 + \frac{\pi B^c_{\mathbf{W}_n}}{\delta^c_{\mathbf{W}\mathbf{W}_n}}\right)\frac{\left(1 + \sqrt{n\log(2n^{3/2})}\right)}{n}\right.$$

$$\left. + \frac{4F^{L-1}L}{n} + 12L^2F^{2L-2}c\right) + \frac{2F^{2L}\sqrt{K}}{\sqrt{n}} \tag{75}$$

*where* $Y_n$ *is the graphon signal induced by* $[\mathbf{y}_n]_i = Y(u_i)$, $u_i = (i - 1)/n$ *for* $1 \le i \le n$ *[cf.* (10)*]. The fixed constants* $B^c_{\mathbf{W}}$ *and* $\delta^c_{\mathbf{W}\mathbf{W}_n}$ *are the c-band cardinality and the c-eigenvalue margin of* $\mathbf{W}$ *and* $\mathbf{W}_n$ *respectively [cf. Definitions 4,5 in the supplementary material].*

*Proof of Theorem 1.* We begin by considering the event $A_n$ such that,

$$A_n = \left(\|\nabla_{\mathcal{H}}\ell(Y, \Phi(X; \mathcal{H}, \mathbf{W})) - \nabla_{\mathcal{H}}\ell(Y_n, \Phi(X_n; \mathcal{H}, \mathbf{W}_n))\|\right. \tag{76}$$

$$\le \sqrt{KF^{L-1}}\left(3L^2F^{2L-2}\left(1 + \frac{\pi B^c_{\mathbf{W}_n}}{\delta^c_{\mathbf{W}\mathbf{W}_n}}\right)\frac{2\left(1 + \sqrt{n\log(\frac{2n}{\xi})}\right)}{n}\right.$$

$$\left.\left. + \frac{4F^{L-1}L}{n} + 12L^2F^{2L-2}c\right)\right).$$

Thus, by considering the disjoint events $A_n$, and $A_n^c$, and denoting $\mathbf{1}(\cdot)$ the indicator function, the expectation can be separated as follows,

$$\mathbb{E}[\|\nabla_{\mathcal{H}}\ell(Y, \Phi(X; \mathcal{H}, \mathbf{W})) - \nabla_{\mathcal{H}}\ell(Y_n, \Phi(X_n; \mathcal{H}, \mathbf{W}_n))\|]$$
$$= \mathbb{E}[\|\nabla_{\mathcal{H}}\ell(Y, \Phi(X; \mathcal{H}, \mathbf{W})) - \nabla_{\mathcal{H}}\ell(Y_n, \Phi(X_n; \mathcal{H}, \mathbf{W}_n))\|\mathbf{1}(A_n)]$$
$$+ \mathbb{E}[\|\nabla_{\mathcal{H}}\ell(Y, \Phi(X; \mathcal{H}, \mathbf{W})) - \nabla_{\mathcal{H}}\ell(Y_n, \Phi(X_n; \mathcal{H}, \mathbf{W}_n))\|\mathbf{1}(A_n^c)] \tag{77}$$

We can bound the term regarding $A_n^c$ using the chain rule, Cauchy-Schwartz inequality, Assumption 4, and Proposition 3 as follows,

$$\|\nabla_{\mathcal{H}}\ell(Y, \Phi(X; \mathcal{H}, \mathbf{W})) - \nabla_{\mathcal{H}}\ell(Y_n, \Phi(X_n; \mathcal{H}, \mathbf{W}_n))\|$$
$$\le \|\nabla_{\mathcal{H}}\ell(Y, \Phi(X; \mathcal{H}, \mathbf{W}))\| + \|\nabla_{\mathcal{H}}\ell(Y_n, \Phi(X_n; \mathcal{H}, \mathbf{W}_n))\| \tag{78}$$
$$\le \|\nabla\ell(Y, \Phi(X; \mathcal{H}, \mathbf{W}))\|\|\nabla_{\mathcal{H}}\Phi(X; \mathcal{H}, \mathbf{W})\|$$
$$+ \|\nabla\ell(Y_n, \Phi(X_n; \mathcal{H}, \mathbf{W}_n))\|\|\nabla_{\mathcal{H}}\Phi(X_n; \mathcal{H}, \mathbf{W}_n)\| \tag{79}$$
$$\le \|\nabla_{\mathcal{H}}\Phi(X; \mathcal{H}, \mathbf{W})\| + \|\nabla_{\mathcal{H}}\Phi(X_n; \mathcal{H}, \mathbf{W}_n)\| \tag{80}$$
$$\le 2F^{2L}\sqrt{K} \tag{81}$$

Returning to equation (77), we can substitute the bound obtained in equation (81), and by taking $P(A_n) = 1 - \xi$, and using Lemma 3, it yields,

$$\mathbb{E}[\|\nabla_{\mathcal{H}}\ell(Y, \Phi(X; \mathcal{H}, \mathbf{W})) - \nabla_{\mathcal{H}}\ell(Y_n, \Phi(X_n; \mathcal{H}, \mathbf{W}_n))\|]$$

$$\le (1 - \xi)\sqrt{KF^{L-1}}\left(3L^2F^{2L-2}\left(1 + \frac{\pi B^c_{\mathbf{W}_n}}{\delta^c_{\mathbf{W}\mathbf{W}_n}}\right)\frac{2\left(1 + \sqrt{n\log(\frac{2n}{\xi})}\right)}{n}\right.$$

$$\left. + \frac{4F^{L-1}L}{n} + 12L^2F^{2L-2}c\right) + \xi 2F^{2L}\sqrt{K} \tag{82}$$

To complete the proof, set $\xi = \frac{1}{\sqrt{n}}$. $\qquad\square$

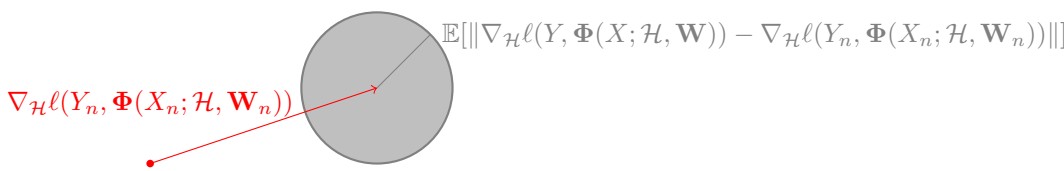

Figure 3: In order to satisfy the property that the inner product between the gradient on the GNN $\nabla_{\mathcal{H}}\ell(Y_n, \mathbf{\Phi}(X_n; \mathcal{H}, \mathbf{W}_n))$ and the gradient on the graphon $\nabla_{\mathcal{H}}\ell(Y, \mathbf{\Phi}(X; \mathcal{H}, \mathbf{W}))$ is positive, we rely on the condition provided in Theorem 2.

## D  PROOF OF THEOREM 2

**Definition 8** (Stopping time). *We define the stopping time $k^*$ as,*

$$k^* = \min_{k \geq 0} \{ \|\nabla_{\mathcal{H}}\mathbf{\Phi}(X; \mathcal{H}_k, \mathbf{W}_n)\| \leq \sqrt{KF^{L-1}}12L^2F^{2L-2}c \}. \tag{83}$$

**Definition 9** (Constant $\psi$).

**Lemma 4.** *Under Assumptions $4, 5$, and $6$, the gradient of the loss function $\ell$ with respect to the parameters of the GNN $\mathcal{H}$ is $A_{\nabla\ell}$-Lipschitz,*

$$\|\nabla_{\mathcal{H}}\ell(Y, \mathbf{\Phi}(X; \mathcal{A}, \mathbf{W})) - \nabla_{\mathcal{H}}\ell(Y, \mathbf{\Phi}(X; \mathcal{B}, \mathbf{W}))\| \leq A_{\nabla\ell}\|\mathcal{A} - \mathcal{B}\| \tag{84}$$

*where $A_{\nabla\ell} = (A_{\nabla\Phi} + A_{\mathbf{\Phi}}F^{2L}\sqrt{K})$.*

*Proof.* To begin with, we can apply the chain rule to obtain,

$$\|\nabla_{\mathcal{H}}\ell(Y, \mathbf{\Phi}(X; \mathcal{A}, \mathbf{W})) - \nabla_{\mathcal{H}}\ell(Y, \mathbf{\Phi}(X; \mathcal{B}, \mathbf{W}))\|$$
$$= \|\nabla\ell(Y, \mathbf{\Phi}(X; \mathcal{A}, \mathbf{W}))\nabla_{\mathcal{H}}\mathbf{\Phi}(X; \mathcal{A}, \mathbf{W}) - \nabla\ell(Y, \mathbf{\Phi}(X; \mathcal{B}, \mathbf{W}))\nabla_{\mathcal{H}}\mathbf{\Phi}(X; \mathcal{B}, \mathbf{W})\| \tag{85}$$

By applying the triangle inequality, and Cauchy-Schwartz it yields,

$$\|\nabla_{\mathcal{H}}\ell(Y, \mathbf{\Phi}(X; \mathcal{A}, \mathbf{W})) - \nabla_{\mathcal{H}}\ell(Y, \mathbf{\Phi}(X; \mathcal{B}, \mathbf{W}))\|$$
$$\leq \|\nabla\ell(Y, \mathbf{\Phi}(X; \mathcal{A}, \mathbf{W}))\|\|\nabla_{\mathcal{H}}\mathbf{\Phi}(X; \mathcal{A}, \mathbf{W}) - \nabla_{\mathcal{H}}\mathbf{\Phi}(X; \mathcal{B}, \mathbf{W})\|$$
$$+ \|\nabla_{\mathcal{H}}\ell(Y, \mathbf{\Phi}(X; \mathcal{A}, \mathbf{W})) - \ell(Y, \mathbf{\Phi}(X; \mathcal{B}, \mathbf{W}))\|\|\nabla_{\mathcal{H}}\mathbf{\Phi}(X; \mathcal{B}, \mathbf{W})\| \tag{86}$$

We can now use Assumptions $1, 4, 5$, and $6$ as well as Proposition 3, to obtain

$$\|\nabla_{\mathcal{H}}\ell(Y, \mathbf{\Phi}(X; \mathcal{A}, \mathbf{W})) - \nabla_{\mathcal{H}}\ell(Y, \mathbf{\Phi}(X; \mathcal{B}, \mathbf{W}))\| \leq (A_{\nabla\Phi} + A_{\mathbf{\Phi}}F^{2L}\sqrt{K}\|\mathcal{A} - \mathcal{B}\|) \tag{87}$$

Completing the proof. $\square$

**Lemma 5.** *Consider the ERM problem in (12) and let $\mathbf{\Phi}(X; \mathcal{H}, \mathbf{W})$ be an L-layer WNN with $F_0 = F_L = 1$, and $F_l = F$ for $1 \leq l \leq L - 1$. Let $c \in (0, 1]$ and assume that the graphon convolutions in all layers of this WNN have $K$ filter taps [cf. (6)]. Let $\mathbf{\Phi}(\mathbf{x}_n; \mathcal{H}, \mathbf{S}_n)$ be a GNN sampled from $\mathbf{\Phi}(X; \mathcal{H}, \mathbf{W})$ as in (9). Under assumptions AS1–AS6, let the following condition be satisfied for every $k$,*

$$\sqrt{KL^{L-1}}\left(6L^2F^{2L-2}\left(1 + \frac{\pi B_{\mathbf{W}_n}^c}{\delta_{\mathbf{W}\mathbf{W}_n}^c}\right)\frac{\left(1 + \sqrt{n\log(2n^{3/2})}\right)}{n} + \frac{4F^{L-1}L}{n} + 12L^2F^{2L-2}c\right)$$
$$+ \frac{2F^{2L}\sqrt{K}}{\sqrt{n}} < \frac{1 - A_{\nabla\ell}\eta_k}{2}\|\nabla_{\mathcal{H}}\ell(Y_n, \mathbf{\Phi}(X_n; \mathcal{H}_k, \mathbf{W}_n))\|. \tag{88}$$

*then the first $k^*$ iterates generated by equation (14), $\mathbf{1}(k \leq k^*)\ell(Y, \mathbf{\Phi}(X; \mathcal{H}_k, \mathbf{W}_n))$ form a positive super-martingale with respect to the filtration $\mathcal{F}_k$ generated by the history of the Algorithm up to step $k$ [i.e., $\{X, Y, X_n, Y_n, \mathbf{W}_n\}_k, \{X, Y, X_n, Y_n, \mathbf{W}_n\}_{k-1}, \ldots, \{X, Y, X_n, Y_n, \mathbf{W}_n\}_0$]. Where $k^*$ is the stopping time defined in Definition 8, and $\mathbf{1}(\cdot)$ is the indicator function.*

*Proof.* To begin with, $\mathbf{1}(k < k^*)\ell(Y, \mathbf{\Phi}(X; \mathcal{H}_k, \mathbf{W})) \in \mathcal{F}_k$, where $\mathcal{F}_k$ is the filtration generated by the history of the Algorithm up to $k$. Note that the loss function $\ell$ is positive by Assumption 4. It remains to be shown the inequality expression of the super-martingale. For $k > k^*$, the inequality is trivially verified as the indicator function $\mathbf{1}(k \leq k^*) = 0$ for $k > k^*$. For $k \leq k^*$, as in Bertsekas & Tsitsiklis (2000), we define a continuous function $g(\epsilon)$ that takes the value of the loss function on the Graphon data on iteration $k + 1$ at $\epsilon = 1$, and on iteration $k$ on $\epsilon = 0$ as follows,

$$g(\epsilon) = \ell(Y, \mathbf{\Phi}(X; \mathcal{H}_k - \epsilon\eta_k\nabla_{\mathcal{H}}\ell(Y_n, \mathbf{\Phi}(X_n; \mathcal{H}_k, \mathbf{W}_n)), \mathbf{W})). \tag{89}$$

Note that function $g(\epsilon)$, is evaluated on the graphon data $Y, X, \mathbf{W}$, but the steps are controlled by the induced graphon data $Y_n, X_n, \mathbf{W}_n$. Applying the chain rule, the derivative of $g(\epsilon)$ with respect to $\epsilon$ can be obtain as follows,

$$\frac{\partial}{\partial\epsilon}g(\epsilon) = \tag{90}$$
$$- \nabla_{\mathcal{H}}\ell(Y, \mathbf{\Phi}(X; \mathcal{H}_k - \epsilon\eta_k\nabla_{\mathcal{H}}\ell(Y_n, \mathbf{\Phi}(\mathcal{H}_k; \mathbf{W}_n; X_n)), \mathbf{W}))\eta_k\nabla_{\mathcal{H}}\ell(Y_n, \mathbf{\Phi}(X_n; \mathcal{H}_k; \mathbf{W}_n)).$$

Now note that the difference in the loss function $\ell$ between iterations $k + 1$ and $k$ can be written as the difference between $g(\epsilon = 1)$ and $g(\epsilon = 0)$ as follows,

$$g(1) - g(0) = \ell(Y, \mathbf{\Phi}(X; \mathcal{H}_{k+1}, \mathbf{W})) - \ell(Y, \mathbf{\Phi}(X; \mathcal{H}_k, \mathbf{W})). \tag{91}$$

Computing the integration of the derivative of $g(\epsilon)$ between $[0, 1]$ it yields

$$\ell(Y, \mathbf{\Phi}(X; \mathcal{H}_{k+1}, \mathbf{W})) - \ell(Y, \mathbf{\Phi}(X; \mathcal{H}_k, \mathbf{W})) = g(1) - g(0) = \int_0^1 \frac{\partial}{\partial\epsilon}g(\epsilon)d\epsilon \tag{92}$$

$$= \int_0^1 \nabla_{\mathcal{H}}\ell(Y, \mathbf{\Phi}(X; \mathcal{H}_k - \epsilon\eta_k\nabla_{\mathcal{H}}\ell(Y_n, \mathbf{\Phi}(X_n; \mathcal{H}_k, \mathbf{W}_n)), \mathbf{W}))$$
$$(-)\eta_k\nabla_{\mathcal{H}}\ell(Y_n, \mathbf{\Phi}(X_n; \mathcal{H}_k, \mathbf{W}_n))d\epsilon. \tag{93}$$

Note that the last term of the previous integral does not depend on $\epsilon$. Besides, we can sum and subtract $\nabla_{\mathcal{H}}\ell(Y, \mathbf{\Phi}(\mathcal{H}_k, \mathbf{W}, X))$ inside the integral, to obtain,

$$\ell(Y, \mathbf{\Phi}(X; \mathcal{H}_{k+1}, \mathbf{W})) - \ell(Y, \mathbf{\Phi}(X; \mathcal{H}_k, \mathbf{W}))$$
$$= (-)\eta_k\nabla_{\mathcal{H}}\ell(Y_n, \mathbf{\Phi}(X_n; \mathcal{H}_k, \mathbf{W}_n))$$
$$\int_0^1 \nabla_{\mathcal{H}}\ell(Y, \mathbf{\Phi}(X; \mathcal{H}_k - \epsilon\eta_k\nabla_{\mathcal{H}}\ell(Y_n, \mathbf{\Phi}(X_n; \mathcal{H}_k, \mathbf{W}_n)), \mathbf{W}))$$
$$+ \nabla_{\mathcal{H}}\ell(Y, \mathbf{\Phi}(X; \mathcal{H}_k, \mathbf{W})) - \nabla_{\mathcal{H}}\ell(Y, \mathbf{\Phi}(X; \mathcal{H}_k, \mathbf{W}))d\epsilon \tag{94}$$
$$= -\eta_k\nabla_{\mathcal{H}}\ell(Y_n, \mathbf{\Phi}(X_n; \mathcal{H}_k, \mathbf{W}_n))\nabla_{\mathcal{H}}\ell(Y, \mathbf{\Phi}(X; \mathcal{H}_k, \mathbf{W}))\int_0^1 d\epsilon$$
$$- \eta_k\nabla_{\mathcal{H}}\ell(Y_n, \mathbf{\Phi}(X_n; \mathcal{H}_k, \mathbf{W}_n))\int_0^1 \nabla_{\mathcal{H}}\ell(Y, \mathbf{\Phi}(\mathcal{H}_k - \epsilon\eta_k\nabla_{\mathcal{H}}\ell(Y_n, \mathbf{\Phi}(X_n; \mathcal{H}_k, \mathbf{W}_n)), \mathbf{W}, X))$$
$$- \nabla_{\mathcal{H}}\ell(Y, \mathbf{\Phi}(X; \mathcal{H}_k, \mathbf{W}))d\epsilon. \tag{95}$$

We can now apply the Cauchy-Schwartz inequality to the last term on the previous inequality, and take the norm of the integral, which is smaller that the integral of the norm to obtain,

$$\ell(Y, \mathbf{\Phi}(X; \mathcal{H}_{k+1}, \mathbf{W})) - \ell(Y, \mathbf{\Phi}(X; \mathcal{H}_k, \mathbf{W}))$$
$$\leq -\eta_k\nabla_{\mathcal{H}}\ell(Y_n, \mathbf{\Phi}(\mathcal{H}_k; \mathbf{W}_n; X_n))\nabla_{\mathcal{H}}\ell(Y, \mathbf{\Phi}(\mathcal{H}_k, \mathbf{W}, X))$$
$$+ \eta_k\|\nabla_{\mathcal{H}}\ell(Y_n, \mathbf{\Phi}(\mathcal{H}_k; \mathbf{W}_n; X_n))\|\int_0^1 \left\|\nabla_{\mathcal{H}}\ell(Y, \mathbf{\Phi}(\mathcal{H}_k - \epsilon\eta_k\nabla_{\mathcal{H}}\ell(Y_n, \mathbf{\Phi}(\mathcal{H}_k; \mathbf{W}_n; X_n)), \mathbf{W}, X))\right.$$
$$\left. - \nabla_{\mathcal{H}}\ell(Y, \mathbf{\Phi}(\mathcal{H}_k, \mathbf{W}, X))\right\|d\epsilon. \tag{96}$$

Under Lemma 4, we can take the Lipschitz bound on the gradient on the loss function with respect to the parameters, using $A_{\nabla\ell}$, to obtain,

$$
\begin{aligned}
&\ell(Y, \mathbf{\Phi}(X; \mathcal{H}_{k+1}, \mathbf{W})) - \ell(Y, \mathbf{\Phi}(X; \mathcal{H}_k, \mathbf{W})) \\
&\leq -\eta_k \nabla_\mathcal{H} \ell(Y_n, \mathbf{\Phi}(X_n; \mathcal{H}_k, \mathbf{W}_n)) \nabla_\mathcal{H} \ell(Y, \mathbf{\Phi}(X; \mathcal{H}_k, \mathbf{W})) \\
&\quad + A_{\nabla\ell} \eta_k \|\nabla_\mathcal{H} \ell(Y_n, \mathbf{\Phi}(X_n; \mathcal{H}_k, \mathbf{W}_n))| \int_0^1 \left\| \eta_k \nabla_\mathcal{H} \ell(Y_n, \mathbf{\Phi}(X_n; \mathcal{H}_k, \mathbf{W}_n)) \right\| \epsilon d\epsilon \quad (97) \\
&\leq -\eta_k \nabla_\mathcal{H} \ell(Y_n, \mathbf{\Phi}(X_n; \mathcal{H}_k, \mathbf{W}_n)) \nabla_\mathcal{H} \ell(Y, \mathbf{\Phi}(X; \mathcal{H}_k, \mathbf{W})) \\
&\quad + \frac{\eta_k^2 A_{\nabla\ell}}{2} \|\nabla_\mathcal{H} \ell(Y_n, \mathbf{\Phi}(X_n; \mathcal{H}_k, \mathbf{W}_n))\|^2. \quad (98)
\end{aligned}
$$

Instead of evaluating the internal product between the gradient on the graphon, and induced graphon, we will use Theorem 1, to bound their expected difference (cf. Figure 3 for intuition). We can add and subtract the gradient of the loss function on the induced graphon $\nabla_\mathcal{H} \ell(Y_n, \mathbf{\Phi}(\mathcal{H}_k; \mathbf{W}_n; X_n))$, and use the Cauchy-Schwartz inequality to obtain,

$$
\begin{aligned}
&\ell(Y, \mathbf{\Phi}(X; \mathcal{H}_{k+1}, \mathbf{W})) - \ell(Y, \mathbf{\Phi}(X; \mathcal{H}_k, \mathbf{W})) \\
&\leq -\eta_k \nabla_\mathcal{H} \ell(Y_n, \mathbf{\Phi}(X_n; \mathcal{H}_k, \mathbf{W}_n)) \\
&\quad (\nabla_\mathcal{H} \ell(Y, \mathbf{\Phi}(X; \mathcal{H}_k, \mathbf{W})) + \nabla_\mathcal{H} \ell(Y_n, \mathbf{\Phi}(X_n; \mathcal{H}_k, \mathbf{W}_n)) - \nabla_\mathcal{H} \ell(Y_n, \mathbf{\Phi}(X_n; \mathcal{H}_k, \mathbf{W}_n))) \\
&\quad + \frac{\eta_k^2 A_{\nabla\Phi}}{2} \|\nabla_\mathcal{H} \ell(Y_n, \mathbf{\Phi}(X_n; \mathcal{H}_k, \mathbf{W}_n))\|^2 \quad (99) \\
&\leq -\eta_k \|\nabla_\mathcal{H} \ell(Y_n, \mathbf{\Phi}(X_n; \mathcal{H}_k, \mathbf{W}_n))\|^2 \\
&\quad + \eta_k \|\nabla_\mathcal{H} \ell(Y_n, \mathbf{\Phi}(X_n; \mathcal{H}_k, \mathbf{W}_n))\| \|\nabla_\mathcal{H} \ell(Y, \mathbf{\Phi}(X; \mathcal{H}_k, \mathbf{W})) - \nabla_\mathcal{H} \ell(Y_n, \mathbf{\Phi}(X_n; \mathcal{H}_k, \mathbf{W}_n))\| \\
&\quad + \frac{\eta_k^2 A_{\nabla\Phi}}{2} \|\nabla_\mathcal{H} \ell(Y_n, \mathbf{\Phi}(X_n; \mathcal{H}_k, \mathbf{W}_n))\|^2. \quad (100)
\end{aligned}
$$

We can rearrange the previous expression, to obtain,

$$
\begin{aligned}
\eta_k \|\nabla_\mathcal{H} \ell(Y_n, \mathbf{\Phi}(X_n; \mathcal{H}_k, \mathbf{W}_n))\|^2 &\left( 1 - \frac{A_{\nabla\ell} \eta_k}{2} \right. \\
&\left. - \frac{\|\nabla_\mathcal{H} \ell(Y, \mathbf{\Phi}(X; \mathcal{H}_k, \mathbf{W})) - \nabla_\mathcal{H} \ell(Y_n, \mathbf{\Phi}(X_n; \mathcal{H}_k, \mathbf{W}_n))\|}{\|\nabla_\mathcal{H} \ell(Y_n, \mathbf{\Phi}(X_n; \mathcal{H}_k, \mathbf{W}_n))\|} \right) \\
&\leq \ell(Y, \mathbf{\Phi}(X; \mathcal{H}_k, \mathbf{W})) - \ell(Y, \mathbf{\Phi}(X; \mathcal{H}_{k+1}, \mathbf{W})). \quad (101)
\end{aligned}
$$

We can now take the conditional expectation with respect to the filtration $\mathcal{F}_n$ to obtain,

$$
\begin{aligned}
&\mathbb{E}[\ell(Y, \mathbf{\Phi}(X; \mathcal{H}_{k+1}, \mathbf{W})) | \mathcal{F}_k] \\
&\leq \eta_k \|\nabla_\mathcal{H} \ell(Y_n, \mathbf{\Phi}(X_n; \mathcal{H}_k, \mathbf{W}_n))\|^2 \left( 1 - \frac{A_{\nabla\ell} \eta_k}{2} \right. \\
&\quad \left. - \mathbb{E}\left[ \frac{\|\nabla_\mathcal{H} \ell(Y, \mathbf{\Phi}(X; \mathcal{H}_k, \mathbf{W})) - \nabla_\mathcal{H} \ell(Y_n, \mathbf{\Phi}(X_n; \mathcal{H}_k, \mathbf{W}_n))\|}{\|\nabla_\mathcal{H} \ell(Y_n, \mathbf{\Phi}(X_n; \mathcal{H}_k, \mathbf{W}_n))\|} \middle| \mathcal{F}_k \right] \right) \\
&\quad + \ell(Y, \mathbf{\Phi}(X; \mathcal{H}_k, \mathbf{W})). \quad (102)
\end{aligned}
$$

As step size $\eta_k > 0$, and by definition norms are non-negative, using Theorem 1, as condition (88) holds for $k \leq k^*$, then

$$
\mathbb{E}[\ell(Y, \mathbf{\Phi}(X; \mathcal{H}_{k+1}, \mathbf{W})) | \mathcal{F}_k] \leq \ell(Y, \mathbf{\Phi}(X; \mathcal{H}_k, \mathbf{W})). \quad (103)
$$

By definition of super-martingale as in Durrett (2019), we complete the proof. $\square$

**Lemma 6.** *Consider the ERM problem in (12) and let $\mathbf{\Phi}(X; \mathcal{H}, \mathbf{W})$ be an L-layer WNN with $F_0 = F_L = 1$, and $F_l = F$ for $1 \leq l \leq L - 1$. Let $c \in (0,1]$ and assume that the graphon convolutions in all layers of this WNN have K filter taps [cf. (6)]. Let $\mathbf{\Phi}(\mathbf{x}_n; \mathcal{H}, \mathbf{S}_n)$ be a GNN*

*sampled from* $\mathbf{\Phi}(X; \mathcal{H}, \mathbf{W})$ *as in (9). Under assumptions AS1–AS6, for any* $\epsilon \in (0, 1 - A_{\nabla \ell}\eta)$, *if the iterates generated by (14), satisfy,*

$$\sqrt{KL^{L-1}}\left(6L^2F^{2L-2}\left(1 + \frac{\pi B^c_{\mathbf{W}_n}}{\delta^c_{\mathbf{WW}_n}}\right)\frac{\left(1 + \sqrt{n\log(2n^{3/2})}\right)}{n} + \frac{4F^{L-1}L}{n} + 12L^2F^{2L-2}c\right)$$
$$+ \frac{2F^{2L}\sqrt{K}}{\sqrt{n}} < \frac{1 - A_{\nabla \ell}\eta_k - \epsilon}{2}\|\nabla_{\mathcal{H}}\ell(Y_n, \mathbf{\Phi}(X_n; \mathcal{H}_k, \mathbf{W}_n)\|. \tag{104}$$

*then the expected value of the stopping time* $k^*$ *[cf. Definition 8], is finite, i.e.,*

$$\mathbb{E}[k^*] = \mathcal{O}(1/\epsilon) \tag{105}$$

*Proof.* Given the iterates at $k = k^*$, and the initial values at $k = 0$, we can express the expected difference between the loss $\ell$, as the summation over the difference of iterates as follows,

$$\mathbb{E}[\ell(Y, \mathbf{\Phi}(X; \mathcal{H}_0, \mathbf{W})) - \ell(Y, \mathbf{\Phi}(X; \mathcal{H}_{k^*}, \mathbf{W}))] =$$
$$\mathbb{E}\left[\sum_{k=1}^{k^*}\ell(Y, \mathbf{\Phi}(X; \mathcal{H}_{k-1}, \mathbf{W})) - \ell(Y, \mathbf{\Phi}(X; \mathcal{H}_k, \mathbf{W}))\right] \tag{106}$$

Taking the expected value with respect to the final iterate $k = k^*$, we get,

$$\mathbb{E}\left[\ell(Y, \mathbf{\Phi}(X; \mathcal{H}_{k^0})) - \ell(Y, \mathbf{\Phi}(X; \mathcal{H}_{k^*}))\right]$$
$$= \mathbb{E}_{k^*}\left[\mathbb{E}\left[\sum_{k=1}^{k^*}\ell(Y, \mathbf{\Phi}(X; \mathcal{H}_{k-1}, \mathbf{W})) - \ell(Y, \mathbf{\Phi}(X; \mathcal{H}_k, \mathbf{W}))\right]\Big|k^*\right] \tag{107}$$
$$= \sum_{t=0}^{\infty}\mathbb{E}\left[\sum_{k=1}^{t}\ell(Y, \mathbf{\Phi}(X; \mathcal{H}_{k-1}, \mathbf{W})) - \ell(Y, \mathbf{\Phi}(X; \mathcal{H}_k, \mathbf{W}))\right]P(k^* = t) \tag{108}$$

Using condition (104), and Lemma 5 for any $k \leq k^*$, it verifies

$$\mathbb{E}\left[\ell(Y, \mathbf{\Phi}(X; \mathcal{H}_{k-1}, \mathbf{W})) - \ell(Y, \mathbf{\Phi}(X; \mathcal{H}_k, \mathbf{W}))\right] \geq \eta(\sqrt{KF^{L-1}}12L^2F^{2L-2}c)^2\epsilon \tag{109}$$

Thus, coming back to (108),

$$\mathbb{E}\left[\ell(Y, \mathbf{\Phi}(X; \mathcal{H}_{k^0}, \mathbf{W})) - \ell(Y, \mathbf{\Phi}(X; \mathcal{H}_{k^*}, \mathbf{W}))\right] \geq \eta(\sqrt{KF^{L-1}}12L^2F^{2L-2}c)^2\epsilon\sum_{t=0}^{\infty}tP(k^* = t) \tag{110}$$
$$\geq \eta(\sqrt{KF^{L-1}}12L^2F^{2L-2}c)^2\epsilon\mathbb{E}[k^*] \tag{111}$$

Note that as the loss function $\ell$ is non-negative,

$$\frac{\mathbb{E}\left[\ell(Y, \mathbf{\Phi}(X; \mathcal{H}_{k^0}, \mathbf{W}))\right]}{\eta(\sqrt{KF^{L-1}}12L^2F^{2L-2}c)^2\epsilon} \geq \mathbb{E}[k^*] \tag{112}$$

Thus concluding that $k^* = \mathcal{O}(1/\epsilon)$. $\square$

**Theorem 2.** *Consider the ERM problem in (12) and let* $\mathbf{\Phi}(X; \mathcal{H}, \mathbf{W})$ *be an L-layer WNN with* $F_0 = F_L = 1$, *and* $F_l = F$ *for* $1 \leq l \leq L - 1$. *Let* $c \in (0, 1]$ *and assume that the graphon convolutions in all layers of this WNN have K filter taps [cf. (6)]. Let* $\mathbf{\Phi}(\mathbf{x}_n; \mathcal{H}, \mathbf{S}_n)$ *be a GNN sampled from* $\mathbf{\Phi}(X; \mathcal{H}, \mathbf{W})$ *as in (9). Consider the iterates generated by equation (14). Under*

*Assumptions AS1-AS6, for any fixed $\epsilon \in (0, 1 - A_{\nabla \ell}\eta)$, if at each step $k$ the number of nodes $n$ is picked such that it verifies*

$$
\sqrt{KL^{L-1}}\left(6L^2F^{2L-2}\left(1 + \frac{\pi B^c_{\mathbf{W}_n}}{\delta^c_{\mathbf{W}\mathbf{W}_n}}\right)\frac{\left(1 + \sqrt{n \log(2n^{3/2})}\right)}{n} + \frac{4F^{L-1}L}{n} + 12L^2F^{2L-2}c\right)
$$

$$
+ \frac{2F^{2L}\sqrt{K}}{\sqrt{n}} < \frac{1 - A_{\nabla \ell}\eta_k - \epsilon}{2}\|\nabla_{\mathcal{H}}\ell(Y_n, \mathbf{\Phi}(X_n; \mathcal{H}_k, \mathbf{W}_n)\|. \tag{113}
$$

*then in finite time we will achieve an iterate $k^*$ such that the coefficients $\mathcal{H}_{k^*}$ satisfy*

$$
\mathbb{E}[\|\nabla_{\mathcal{H}}\ell(Y, \mathbf{\Phi}(X; \mathcal{H}_{k^*}, \mathbf{W}))\|] \leq 24\sqrt{KF^{L-1}}L^2F^{2L-2}c \quad \text{with probability } 1 \tag{114}
$$

*where $A_{\nabla \ell \eta_k} = (A_{\nabla \Phi} + A_{\mathbf{\Phi}}F^{2L}\sqrt{K})$.*

*Proof.* We can use Lemma 6, to conclude that it must be the case that $P(k^* = \infty) = 0$, which implies that, $P(k^* < \infty) = 1$. Using stopping time $k^*$ condition [cf. Definition 8] and the triangle inequality, it yields,

$$
\mathbb{E}[\|\nabla_{\mathcal{H}}\ell(Y, \mathbf{\Phi}(X; \mathcal{H}_{k^*}, \mathbf{W}))\|] \leq \|\nabla_{\mathcal{H}}\ell(Y_n, \mathbf{\Phi}(X_n; \mathcal{H}_{k^*}, \mathbf{W}_n))\| \tag{115}
$$
$$
+ \mathbb{E}[\|\nabla_{\mathcal{H}}\ell(Y_n, \mathbf{\Phi}(X; \mathcal{H}_{k^*}, \mathbf{W}_n)) - \nabla_{\mathcal{H}}\ell(Y, \mathbf{\Phi}(X_n; \mathcal{H}_{k^*}, \mathbf{W}_n))\|]
$$

Note that the iterates are constructed such that, for every $k$

$$
\mathbb{E}[\|\nabla_{\mathcal{H}}\ell(Y_n, \mathbf{\Phi}(X; \mathcal{H}_k, \mathbf{W}_n)) - \nabla_{\mathcal{H}}\ell(Y, \mathbf{\Phi}(X_n; \mathcal{H}_k, \mathbf{W}_n))\|] \leq \|\nabla_{\mathcal{H}}\ell(Y_n, \mathbf{\Phi}(X; \mathcal{H}_k, \mathbf{W}_n))\|. \tag{116}
$$

Using the stopping time condition, the final result is attained as follows

$$
\mathbb{E}[\|\nabla_{\mathcal{H}}\ell(Y, \mathbf{\Phi}(X; \mathcal{H}_{k^*}, \mathbf{W}))\|] \leq 2\|\nabla_{\mathcal{H}}\ell(Y_n, \mathbf{\Phi}(X; \mathcal{H}_{k^*}, \mathbf{W}_n))\| \tag{117}
$$
$$
\leq 24\sqrt{KF^{L-1}}L^2F^{2L-2}c. \tag{118}
$$

$\square$

