# OpenReview forum: "Increase and Conquer: Training Graph Neural Networks on Growing Graphs"
_ICLR.cc/2022/Conference — ICLR 2022 Submitted_

### Official Review · Reviewer_3Wdv · 2021-10-26

**Correctness:** 3
**Technical Novelty And Significance:** 3
**Empirical Novelty And Significance:** 2
**Recommendation:** 3
**Confidence:** 4

**Main Review:**

**Strengths**

1. The paper is easy to follow, and it does a good job of describing the WNN proposed by Ruiz et al., 2020. The introduction to GNNs is similarly clear.

2. The poor scalability of GNNs is an important problem that the community should solve to facilitate the adoption of GNNs in large-scale settings. This work is an attempt to address that problem.

3. The authors do a good job of explaining the theoretical results intuitively, making it easy for the reader to navigate the somewhat complex math derivations.

**Weaknesses**

1. Theorem 1 states a similar thing to Theorem 1 of Ruiz et al., 2020: the distance (in one case of the output, in the other case of the gradients) between the GNN and the WNN is bounded by a term that depends on the number of nodes and a few other hyperparameters.
The authors themselves point this out in Section 2, saying that their results are an "extension" of the other paper's.

    - Why are the differences between the two theorems significant? Why couldn't the results of Theorem 2 be expressed in light of Theorem 1 by Ruiz et al.?

    - There are also some similarities between the two papers, down to things like notation, (sub)section names, the content of some sections, and one of the two experiments. The authors should highlight better this "debt" towards the other paper.

2. How does the theory guide the user in choosing how many nodes to add at each epoch? The authors compare adding 2.5%, 5%, 7.5% and 10% of the full dataset at each epoch: how were these numbers chosen? Isn't the cost reduced only by a constant factor and only for the first few epochs?

    Related to this: it is not clear if the authors want to improve the total training time or the maximum cost in terms of memory/size, and they should make it clearer.
    In either case, anyway, the algorithm is only a temporary stall that ultimately has the same scalability issues as the typical training procedure.

3. When using real graphs, there may not be enough nodes to satisfy (16) and therefore convergence is not guaranteed.
    - What happens in that case?
    - How soon does $k^*$ occur, in practice, when using Algorithm 1?
    - What is the minimum number of nodes, in practice, that we can add at each iterate while ensuring that the GNN follows the WNN?

    The authors should perform a controlled/synthetic experiment to give a quantitative answer to these questions, which would also alleviate the concerns I raised in the previous comments and provide valuable insight to the reader.

4. How does the proposed method compare to typical subgraph batching?
By training on random subgraphs, the cost is fixed and the GNN is exposed to a great diversity of local neighbourhoods structures.
This is an essential baseline that cannot be ignored in the analysis of Algorithm 1 (since it also has a lower overall cost than what was proposed by the authors).

5. One of the main objectives of the paper is to "address the computational burden of training a GNN on a large graph".
However, the largest graph used in the experiments has 1000 nodes while the other one has 100 nodes.
The experiments should be run on one or more large-scale graphs to empirically show that the algorithm works (e.g., some OGB benchmark).

6. Figure 1 seems to indicate that there is no difference in how many nodes are added at each epoch, and that having 200-300 nodes is already more than enough for achieving performance similar to when we have 1000 nodes.
This seems like a result about the data efficiency of GNNs more than a result about Algorithm 1.

7. Figure 2 should show the convergence of the GNN trained on the full graph.
It could be that the full GNN converges quickly to a low error, meaning that we could stop training much earlier than 10 epochs.
If one of the advantages of the proposed method is to reduce the computational cost of training the GNN, then this advantage should be quantified in the experiments.

**Other comments**

8. Figure 2 can confuse the reader because the red and purple horizontal lines are not a function of the epochs, they indicate a constant value.

9. It would have been nice if the authors had included the code to reproduce their experiments with the submission.

10. Sec. 3.1, first paragraph: the citation to Gama et al. 2018 uses the textual format instead of the parenthetical. Also, I'm not sure that a GNN paper from 2018 is the best reference for basic notions of graph theory like the adjacency matrix and the Laplacian.

11. Line 4 of Algorithm 1 assumes that a dataset of graphons is available. This is never the case, so that line/notation should be changed or improved.

**References**

Ruiz et al., "Graphon neural networks and the transferability of graph neural networks", Neural Information Processing Systems (2020).

**Summary Of The Paper:**

This paper shows that a graph neural network (GNN) trained on an increasingly larger graph behaves like a graphon neural network (WNN), the "limit object of a GNN", which arises when considering a continuous graphon instead of a discrete graph (the graphon can be seen as a generative model from which the discrete graph is sampled).

This observation is translated into two theorems that bound the expected distance between the GNN and the WNN as a function of the number of nodes used to train the GNN.

The authors then propose an algorithm to train a GNN by gradually adding nodes to the training data, leading to a computational advantage compared to training the GNN on the full dataset from the start.

**Summary Of The Review:**

In its current state, the paper is not ready for publication:

- The relation with Theorem 1 of Ruiz et al. 2020 is unclear.
- The theoretical analysis is not sufficient to justify Algorithm 1, since it is based on an assumption that might not hold in practice.
- The paper does not show evidence of having solved the main problem that it sets out to solve (ie, reducing computational costs).
- The experimental evaluation is insufficient (not enough experiments, no baselines, and too small graphs) and the results raise some questions.

---

### Official Review · Reviewer_TUDu · 2021-11-02

**Correctness:** 3
**Technical Novelty And Significance:** 3
**Empirical Novelty And Significance:** 3
**Recommendation:** 6
**Confidence:** 4

**Main Review:**

Pros:
1) Reducing the computational burden of training a GNN is important, as in practical use cases, e.g., graphs like social networks usually are huge, and cost many resources to train and fine-tune.
2) Proofs on the feasibility of the proposed method and convergence of algorithms are of great interest and give an intuition for why an algorithm would work.
3) In general, the main flow of the paper and ideas are clear.

Cons/questions needed to be addressed:
1) I observed that $F_0=1$ is highlighted many times in the paper, which means that the input is $\bf{x}\in \mathbb{R}^n$. Does it mean that the derived proof and conclusion only apply to the case where the input feature at each node has dimension one? Please clarify this.
2) In Theorem 1 and 2, please mention the definition of $\bf{W}_n$ or $\bf{\Phi}(\bf{x}_n; \mathcal{H}, \bf{W}_n$) to make them self-contained.
3) Some comments should be added to clarify how Theorem 1 and 2 connect to GNN, i.e., $\bf{\Phi}(\bf{x}_n; \mathcal{H}, \bf{S}_n)$. The proofs are done for $\bf{\Phi}(\bf{x}_n; \mathcal{H}, \bf{W}_n)$, but to my understanding, Algorithm 1 produces $\bf{\Phi}(\bf{x}_n; \mathcal{H}, \bf{S}_n)$ (or $\bf{S}_n$). Are $\bf{\Phi}(\bf{x}_n; \mathcal{H}, \bf{S}_n)$ and $\bf{\Phi}(\bf{x}_n; \mathcal{H}, \bf{W}_n)$ asymptotically close to each other in some sense?
4) For theorem 1, Is (15), i.e., that the learning steps of $\bf{\Phi}(\bf{x}_n; \mathcal{H}, \bf{W}_n)$ and $\bf{\Phi}(\bf{x}_n; \mathcal{H}, \bf{W})$ are close, sufficient to indicate the $\bf{W}$ and $\bf{S}_n$ are "close" in some sense? Please clarify this.
5) In Section 5, Figure 1(b). It seems that the "25 Nodes Added (blue line)" case has the best performance as the relative RMSE drops at epoch 2. Is this expected? Further, please give some intuitions/or suggestions for how to choose the number of nodes added in each step.
6. It could be future work, but to me, the graph sizes in the numerical results are not very large. It would be great to see the results on larger graph sizes, so that the contributions of this paper are well supported.

A minor issue:
1) Before (10), "a WNN ca be" should be "a WNN can be".

**Summary Of The Paper:**

In this paper, the authors propose a method that learns a large graph neural network (GNN) starting from a relatively smaller GNN which would increase its number of nodes step by step (epoch by epoch). Most importantly, the paper gives mathematical proof to show that the gradient descent steps on GNN are close to the graphon neural network (WNN) which is the limit object of the GNN. This paper also demonstrates its proposed method on a recommendation system and a decentralized control problem and shows reduced computational cost.

**Summary Of The Review:**

In general, I think the problem investigated in the paper is important, and learning from smaller GNN to approximate its limit object is of interest. Also, it is great to see the theorems and proofs are well stated in the paper to help understand the algorithm. However, I still hope that the authors could answer the question or make clarifications to some issues mentioned above.

---

### Official Review · Reviewer_n2bA · 2021-11-02

**Correctness:** 2
**Technical Novelty And Significance:** 3
**Empirical Novelty And Significance:** 3
**Recommendation:** 3
**Confidence:** 4

**Main Review:**

Pros.
1.	The writing of this paper is clear. It is easy to follow the idea and algorithm details proposed by the authors.
2.	The theoretical guarantee of the algorithm is solid.
3.	The convergence performance on the small datasets in the experimental part seems effective.

Cons.
1.	Motivation. This paper seems to lack a good elaboration of the motivation on why using graphon neural networks. The authors claim in Section 3.2.2 that graphon neural networks can be seen as a generative model for GNN. Can we understand that the graphon neural network is only used to generate a small subgraph from the original graph? If yes, then it would be better for the authors to further explain why other simple generative methods are not chosen as the way to generate the subgraph. If not, can the author further explain what are the superior characteristics of applying graph neural networks?
2.	Previous work. I think some previous works [1,2,3] were also proposed to address the issue of scalability of training a graph neural network model. How is the proposed method compared to them?
3.	Experiments. The settings for the experiments are not convincing to me. First, the authors claim that their methods are designed to handle the issue of the scalability of training graph neural networks. However, the size of datasets in the experiments is only ~1k. I think it would be better to conduct some experiments on significantly large datasets. Second, in Fig 2 the authors only show the results of training a normal graph neural network by 10 epochs and 30 epochs. There is no evidence that the training of the graph neural network has already converged by this epoch. As a result, it is not convincing that the algorithm proposed can converge to a normally trained graph networks model. Third, there is no result of previous working methods.
4.	Further discussion of the universality of the proposed model is important. Does this method fit for all kinds of graph neural networks or only some specific models?



[1] Chiang, Wei-Lin, et al. "Cluster-GCN: An efficient algorithm for training deep and large graph convolutional networks." Proceedings of the 25th ACM SIGKDD International Conference on Knowledge Discovery & Data Mining. 2019.
[2] Zeng, Hanqing, et al. "Graphsaint: Graph sampling-based inductive learning method." arXiv preprint arXiv:1907.04931 (2019).
[3] Chen, Jie, Tengfei Ma, and Cao Xiao. "FastGCN: Fast Learning with Graph Convolutional Networks via Importance Sampling." International Conference on Learning Representations. 2018.


**Summary Of The Paper:**

This paper proposed an effective and scalable algorithm to train the graph neural network. Their method leverages the framework of graphon neural networks to enlarge the training size of GNN during the training. Specifically, a growing size subgraph is sampled from graphon by a Bernoulli distribution and fed to a graph neural network. A Theoretical guarantee is given for the convergence of the algorithm by proving the absolute value of gradient will decrease to a small value surely.

**Summary Of The Review:**

In conclusion, this paper proposed a novel way to handle the issue of scalability of training a graph neural network model. However, the disadvantages of the paper mentioned above, especially the lack of a well-elaborated motivation and the weakness of experimental settings, hurt the value of this work. Thus, a reject is recommended here.

---

### Decision · Program_Chairs · 2022-01-20

**Decision:**

Reject

**Comment:**

This paper proposes a scalable learning method for GNN that gradually increases the training data size by randomly adding vertexes generated from a graphon. Theoretical justification to the proposed method is given that bounds the difference between the gradients on the sampled network and on the graphon. A numerical experiment was conducted to support the validity of the proposed method.

Unfortunately, this paper contains several issues as listed below:
1. Novelty: There are already some existing work  to address the issue of scalability of training a graph neural network model. However, the relation to them is not appropriately exposed.
2. Experiments: Although the main purpose of this paper is to resolve the scalability of GNN, the numerical experiments are conducted only on a small scale dataset ($\sim$1k).
3. Practicality: There are several hyperparameters. However, the theory and methodology do not give a practical guideline to determine them (e.g., how many vertexes should be added at each epoch).
4. Correctness: The proof of the theorems would contain some flaws, which should be resolved by the authors. However, there was no response from the authors.

For these reasons, this paper would not be appropriate to appear in ICLR.